# Multiomic profiling of chronically activated CD4+ T cells identifies drivers of exhaustion and metabolic reprogramming

**Matthew L. Lawton** [1,2], **Melissa M. Inge** [3], **Benjamin C. Blum** [1,2], **Erika L. Smith-Mahoney** [4], **Dante Bolzan** [5], **Weiwei Lin** [1,2], **Christina McConney** [4], **Jacob Porter** [6], **Jarrod Moore** [1,2], **Ahmed Youssef** [1,2], **Yashasvi Tharani** [1,3], **Xaralabos Varelas** [2], **Gerald V. Denis** [7], **Wilson W. Wong** [8], **Dzmitry Padhorny** [9,10], **Dima Kozakov** [9,10], **Trevor Siggers** [3], **Stefan Wuchty** [5,11], **Jennifer Snyder-Cappione** [4], **Andrew Emili** [1,2,3,6]*

1 Center for Network Systems Biology, Boston University School of Medicine, Boston, Massachusetts, United States of America, 2 Department of Biochemistry, Boston University School of Medicine, Boston, Massachusetts, United States of America, 3 Depart of Biology, Boston University, Boston, Massachusetts, United States of America, 4 Department of Microbiology, Boston University School of Medicine, Boston, Massachusetts, United States of America, 5 Department of Computer Science, University of Miami, Miami, Florida, United States of America, 6 Knight Cancer Institute, Oregon Health and Science University, Portland, Oregon, United States of America, 7 Hematology and Medical Oncology, Boston University School of Medicine, Boston, Massachusetts, United States of America, 8 Department of Biomedical Engineering, Boston University, Boston, Massachusetts, United States of America, 9 Department of Applied Mathematics and Statistics, Stony Brook University, Stony Brook, New York, United States of America, 10 Laufer Center for Physical and Quantitative Biology, Stony Brook University, Stony Brook, New York, United States of America, 11 Miami Institute of Data Science and Computing, Miami, Florida, United States of America

* emili@ohsu.edu

**Data Availability Statement:** All relevant data and supplemental data are within the paper and its Supporting Information files excluding raw flow cytometry and mass spectrometry files, which can

## Abstract

Repeated antigen exposure leads to T-cell exhaustion, a transcriptionally and epigenetically distinct cellular state marked by loss of effector functions (e.g., cytotoxicity, cytokine production/release), up-regulation of inhibitory receptors (e.g., PD-1), and reduced proliferative capacity. Molecular pathways underlying T-cell exhaustion have been defined for CD8+ cytotoxic T cells, but which factors drive exhaustion in CD4+ T cells, that are also required for an effective immune response against a tumor or infection, remains unclear. Here, we utilize quantitative proteomic, phosphoproteomic, and metabolomic analyses to characterize the molecular basis of the dysfunctional cell state induced by chronic stimulation of CD4 + memory T cells. We identified a dynamic response encompassing both known and novel up-regulated cell surface receptors, as well as dozens of unexpected transcriptional regulators. Integrated causal network analysis of our combined data predicts the histone acetyltransferase p300 as a driver of aspects of this phenotype following chronic stimulation, which we confirmed via targeted small molecule inhibition. While our integrative analysis also revealed large-scale metabolic reprogramming, our independent investigation confirmed a global remodeling away from glycolysis to a dysfunctional fatty acid oxidation-based metabolism coincident with oxidative stress. Overall, these data provide both insights into the mechanistic basis of CD4+ T-cell exhaustion and serve as a valuable resource for future interventional studies aimed at modulating T-cell dysfunction.

be found in the following locations: Flow cytometry files are freely accessible at http://flowrepository.org using the identifier "FR-FCM-Z8G6", and the mass spectrometry data files have been deposited to the ProteomeXchange Consortium via the PRIDE partner repository with the dataset identifier PXD057703. For more information, please see reference below. Perez-Riverol Y, Bai J, Bandla C, Hewapathirana S, García-Seisdedos D, Kamatchinathan S, Kundu D, Prakash A, Frericks-Zipper A, Eisenacher M, Walzer M, Wang S, Brazma A, Vizcaíno JA (2022). The PRIDE database resources in 2022: A Hub for mass spectrometry-based proteomics evidences. Nucleic Acids Res 50 (D1):D543-D552 (PubMed ID: 34723319).

**Funding:** This work was funded by the following grants: U01CA243004, National Cancer Institute, https://www.cancer.gov/ to G.D., A.E. R01 AI151051, National Institute of Allergy and Infectious Diseases, https://www.niaid.nih.gov/ to T.S. The funders did not play a role in study design, data collection and analysis, decision to publish, nor preparation of the manuscript.

**Abbreviations:** ACT, adoptive cell transfer; CAR, chimeric antigen receptor; FACS, fluorescence-activated cell sorting; GSEA, Geneset Enrichment Analysis; IC, information content; IR, inhibitory receptor; LCMV, lymphocytic choriomeningitis viral; LLE, liquid–liquid extraction; MFI, mean fluorescent intensity; NES, normalized enrichment score; nLC-MS, nano-liquid chromatography mass spectrometer; PBM, protein-binding microarray; PBMC, peripheral blood mononuclear cell; PRC2, polycomb repressive complex 2; PWH, people living with HIV; PWM, position weight matrix; ROS, reactive oxygen species; RT, room temperature; SLAM, signaling lymphocytic activation molecule; SPME, solid phase micro-extraction; SV, single variant; TFBS, transcription factor binding site; Tfh, T follicular helper; Th, T helper; VDR, vitamin D receptor; VEGFR, vascular endothelial growth factor receptor.

## Introduction

T-cell exhaustion is an important phenotypic manifestation that results from repeated T-cell stimulation with broad pathobiological relevance. Originally identified in chronic lymphocytic choriomeningitis viral (LCMV) infections in mice [1–3], this dysfunctional cellular state typically occurs when T cells are continuously exposed to persistent antigens. Both in vitro and in vivo, chronic T-cell activation leads to eventual loss of key effector functions, including cytotoxicity and cytokine secretion, a loss of proliferative capacity, and an up-regulation of multiple inhibitory receptors such as PD-1 [4], LAG-3 [5], TIM-3 [6], and TIGIT [7] (also referred to as immune checkpoint proteins), with the full details of exhaustion reviewed elsewhere [8–10]. T-cell exhaustion and chronic antigen exposure has also been associated with changes in cellular metabolism. Exhausted/chronically stimulated immune cells show a characteristic shift in metabolism away from glycolysis in favor of fatty acid oxidation as energy source, in conjunction with increased oxidative stress and mitochondrial dysfunction [11,12]. However, it remains unclear how the cellular context (i.e., tumor microenvironment) and resulting lack of nutrients induces or interplays with these observed metabolic changes.

This phenotype has profound clinical implications, leading to ineffectual pathogen and tumor clearance. For example, checkpoint blockade therapies have seen success through improvement of immune responses to tumors by blocking inhibitory receptors expressed on exhausted T cells [13–15]. Genetically modified adoptive cell transfer (ACT) therapies, most notably chimeric antigen receptor (CAR) T cells, aim to reprogram native immunity to increase effector functions and delay exhaustion [16–18], as reviewed in many places [19–21]. However, even with the clinical impact seen with checkpoint blockade for some cancers, only 15% to 60% of patients with solid tumors respond [22]. Therefore, the effectiveness of immunotherapies would benefit from a more complete understanding of T-cell exhaustion following chronic antigen exposure. Thorough knowledge of mechanisms that underly T-cell exhaustion is also of interest in other clinically relevant areas where T cells are often hyperactive such as autoimmune disorders, as it may be beneficial to induce a subdued exhaustion-like state in these contexts. While certain markers and drivers of T-cell exhaustion seen in tumor infiltrates (TILS) have been well studied in the past decade [8,10,23,24], most findings are limited to CD8 + subtypes, leaving unanswered questions if these observations carry over to their CD4+ T-cell counterparts.

CD4+ T cells serve several key roles in adaptive immunity. While it is important to understand exhaustion in cytotoxic effector cells, as they are responsible for a large part of the direct cytotoxicity within a tumor, it is becoming increasingly apparent that exhaustion of CD4+ T cells likewise plays a critical role in anti-tumor immunity. T helper (Th) cells, are classified into 5 main subsets (Th1, Th2, Th17, T follicular helper (Tfh), and T regulatory (T_reg) cells), control important immune functions including secreting cytokines that signal various responses, regulating inflammation, licensing dendritic cells to prime CD8+ T cells, signaling B cells for antibody class switching, recruiting other immune cells such as neutrophils, influencing angiogenesis, and cytotoxicity [25–31]. Increased presence of CD4+ T cells is correlated with better survival in cancer patients [32,33], while a higher CD4-to-CD8 ratio of T cells in leukapheresis products used for making CAR T therapies has been shown to improve anti-tumor responses with better clinical outcomes [30,34,35]. CD4+ T cells serve several key roles in adaptive immunity, while CD4+ CAR T cells have been reported in some cases to have equal killing capacity compared to CD8+ CAR T cells [36]. CD4+ T cells have also been shown in some cases to be the main responder during therapies like neoantigen vaccinations [37,38].

During infection/challenge, naïve CD4+ T cells differentiate into mature, effector cells which function to clear the challenge, then subsequently differentiating further to memory

CD4+ cells. In particular, CD4+ memory cells have strong immunity upon rechallenge, with efficient effector functions and proliferation. Upon repeated antigen exposure, however, CD4+ memory cells lose their functionality and display much of the same phenotype to CD8+ T cells, as excellently reviewed by Miggelbrink and colleagues [27]. Specifically, up-regulation of inhibitory receptors on CD4+ cells is observed in the classic LCMV model [39,40], which is often used for chronic infection and exhaustion studies. Similar to CD8+ T-cell exhaustion, CD4+ T cells are known to up-regulate the same inhibitory receptors, including PD-1, TIM-3, LAG-3, and TIGIT [41]. In addition, such cells show higher expression of these receptors when chronically exposed to antigen [42], which is correlated with higher disease severity with both infections and tumors [6,43–46]. Furthermore, CD4+ T cell from diseased or infected tissues have been shown to express increased levels of these markers [42,47], suggesting similar responses to CD8+ T cells during repeated antigen exposure. However, it is not enough to demark exhaustion from inhibitory receptor expression alone, as these proteins can also be up-regulated during activation. In addition to inhibitory receptor expression, reduced function is also observed. This includes impaired cytokine secretion [43,48], reduced splenic mobility [49], and reduced response to secondary challenge [48]. For example, a study investigating the effects of chronic antigen stimulation in mouse tumors reported reduced CD4+ T-cell proliferation, cytokine production, and antitumor responses [50]. While exhaustion hallmarks of increased inhibitory receptor expression and loss of function are observed in CD4+ T cells, suggesting a similar exhausted phenotype, there is evidence that not all effector function is lost. It has been reported that chronically exposed CD4+ T cells increase expression of granzyme B [51] and perforin [52], a slight regain of function which is also seen in terminally exhausted CD8+ T cells [53]. Investigations of the transition from progenitor to terminal exhausted state of CD4+ T cells found down-regulation of naïve markers (TCF1, SLAMF6) on tumor-infiltrating CD4+ T cells compared to cells in the spleen [54], leading to ACT failure that could be rescued with checkpoint blockade (anti-PD-L1 treatment). While critical to effective anti-tumor responses, key aspects of CD4+ T-cell exhaustion remain unclear. Key unknowns include the signaling mechanisms, metabolic changes and their causes, possible subtype-specific markers, and extent of the terminal phenotype(s) of exhausted CD4+ T cells as compared to CD8+ T cells. Central to addressing this gap is a better understanding of the main molecular pathways driving CD4+ T cells to an exhausted state and fully characterizing protein expression, signaling pathways, and metabolism of these cells.

To determine the defining characteristics of CD4+ T-cell exhaustion with increased scope, we performed a comprehensive multiomic time course analysis of human primary CD4+ T cells that had been chronically stimulated in vitro using an established culture model system [55–59]. Our systematic, longitudinal characterization of the cultured CD4+ T cells elucidated the dynamics of a composite CD4+ memory T-cell response to repeated antigen simulation. In addition to measuring cell-surface receptors, cell proliferation, and cytokine secretion, we recorded quantitative bottom-up proteomic profiles of global protein expression as CD4+ T-cells transition from late activation to exhaustion, together with phosphoproteomics to illuminate the signaling pathways driving this transition. Integrative network analyses were able to identify drivers of exhaustion, including the histone acetyltransferase p300, which when inhibited was confirmed to regulate aspects of T-cell exhaustion such as inhibitory receptor expression (i.e., PD-1, TIM-3). Our data also revealed large-scale metabolic remodeling that is distinct from observations in CD8+ cells, which we confirmed using untargeted metabolomics. Collectively, our results establish novel components of the dynamic pathway responses leading to phenotypic dysfunction in CD4+ T cells.

## Results

### CD4+ memory T cells exhibit progressive exhaustion phenotype during chronic stimulation in vitro

While a complex phenotype in the context of a tumor or infection, T-cell exhaustion can be replicated in culture models by repeatedly exposing primary T cells to antigen or CD3/CD28 activation [55], and has been used for discovery of mediators of the exhausted state [56–59]. We adapted this protocol to profile the molecular kinetics of human CD4+ T-cell exhaustion, focusing on memory CD4+ T cells to minimize cellular heterogeneity, given their abundance and critical functions in the tumor microenvironment. We chose the memory subset not only for their functionality in anti-tumor responses, but also as a starting cell population that would allow us a balance of specificity/homogeneity (versus whole CD4+ T cell population) and practicality (allowing us to isolate a large enough population to culture to perform our experiments). These cells also expand well in response to antigen stimulation, a benefit for our downstream analysis material needs.

To define the molecular basis of dynamic cellular state transitions of the initiation and progression of T-cell exhaustion, our pipeline employed multi-pronged quantitative proteomic, phosphoproteomic, and metabolomic workflows to unbiasedly analyze changes in global protein expression, phosphorylation, and metabolite abundance. CD4+ CD45RO+ T cells were isolated from 2 healthy donors (donor #1 primarily shown, but donor #2 expression and protein agreement can be found in **S1 Fig**. Complete data set in **S1 Data**). Donor #2 samples were collected at slightly differing time points from donor #1 to capture different expression dynamics during chronic stimulation, but the key time points of 0, 6, and 12 days were kept the same for comparison between data sets. T cells were expanded prior to start of the chronic stimulation protocol, whereby cells were next subjected to chronic CD3 and CD28 stimulation using magnetic bead-bound antibodies in the presence of low levels of IL-2 (to maintain cell viability, **S2 Fig**) over the course of 12 days (**Fig 1A**). Media and antigen (CD3/CD28 Dynabeads) were replenished every 2 days, and cell aliquots were collected for phenotyping, cytokine profiling, and multiomic analysis at select intervals (**Fig 1B**). We profiled cell cultures harvested at representative time points (days 6, 8, 10, and 12), along with cells sustained in a "resting" condition (i.e., stimulation for 2 days before antigen removal followed by culturing without antigen until day 10 in cultures parallel to those undergoing repeated stimulation) (**Fig 1A**).

Conditioned media was recovered at each time point for 3 prominently secreted cytokines (IL-2, IFN-γ, TNF-α) 8 to 10 h after re-stimulation to confirm decreased secretion as typically seen in exhaustion. Consistent with previous studies [55,56], we observed a marked decrease in cytokine secretion by 10 days with almost complete loss of these functions by 12 days (**Figs 1C and S3**), along with decreased cell proliferation (**Fig 1D**), and increased expression of well-known inhibitory receptors (PD-1, TIM-3, LAG-3, TIGIT) (**Figs 1E and S1A**). IL-2 secretion was down-regulated earliest, reaching almost undetectable levels by day 6, whereas IFN-γ and TNF-α showed maximally reduced levels by day 12 (**Fig 1C**). Cell proliferation followed a similar trajectory, with high cell division until day 10, followed by decreased cell division rates until proliferation became stagnant by day 12 (**Fig 1D**). Some cell death was observed throughout the protocol, especially in the early stages of stimulation (day 2 through day 6 or 8), leveling out later in the protocol (day 8 to 10), and slightly increasing by day 12 as the cells enter later stages of chronic stimulation. As expected, flow cytometry measurements revealed pronounced up-regulation of cell surface levels of PD-1, TIM-3, TIGIT, LAG-3, CD160, and CD137 (**Figs 1E and S4**), with some level of heterogeneity of marker co-expression (**S4 Fig**).

Taken together, these phenotypic results indicated that the CD4+ T cells were functionally exhausted by day 12. With confidence in our model, we went forward with our systematic

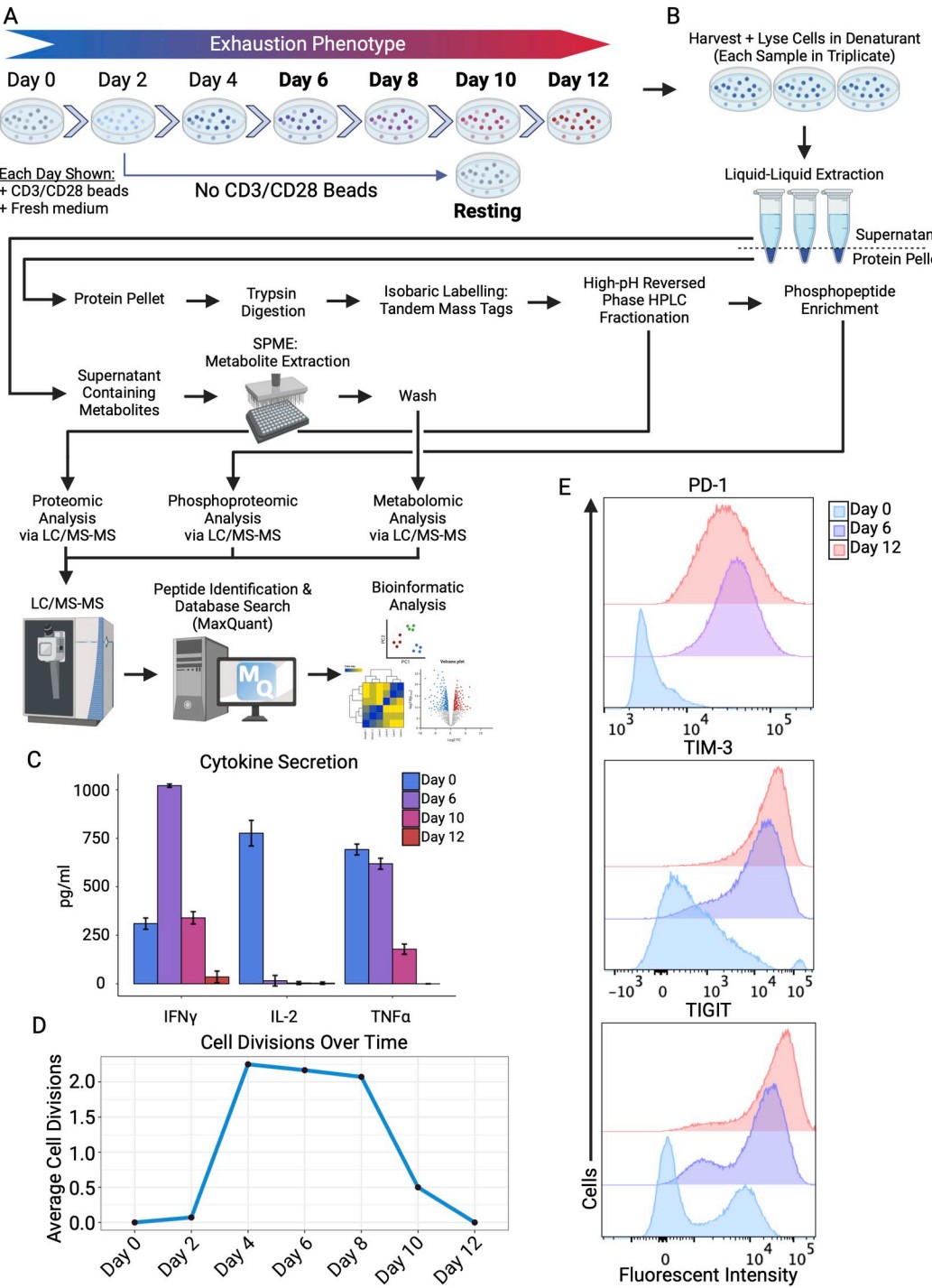

**Fig 1. In vitro chronic CD4+ T-cell activation model replicates exhaustion phenotype while providing material for multiomics workflow.** (A) Diagram depicting the culture model used to generate time course samples from healthy human CD4+ memory T cells. Cells were stimulated with CD3/CD28 Dynabeads for 12 days, with fresh beads plus full media change every other day (fresh media alone added on alternate days). Samples were collected on days highlighted in bold. (B) Multiomics workflow (see Methods for details). Experiments were performed in biological triplicate, with proteomics, phosphoproteomics, and metabolomics done on same samples. (C) ELISA-based quantification of secreted cytokines (IL-2, IFN-γ, TNF-α) on days 0, 6, 10, and 12, 3 replicates per time point. (D) Cell counts were taken each time point, and average cell division compared to previous time point calculated. Cell viability remained stable throughout at ~75%–85%. (E) Flow cytometric analysis of inhibitory receptors PD-1, TIM-3, and TIGIT at resting, day 6, and day 12 time points. Data normalized to mode for accurate population comparisons to account for differences in number of cells analyzed. The data

underlying Fig 1E can be found in the supplemental flow cytometry files uploaded to flowrepository.org. Figure created with BioRender.com.

characterization of samples collected over a time course, allowing us to capture how CD4 + memory T cells respond to chronic antigen simulation.

## Quantitative proteomic analysis of chronically stimulated CD4+ memory cells illuminates temporal signaling events associated with exhaustion

Utilizing quantitative mass spectrometry using isobaric stable-isotope-based multiplex labeling (TMT), we measured dynamic changes in protein expression during the exhaustion time course. Post-filtering and normalization (S5A Fig), we detected and quantified with high confidence changes in the relative abundance a total of 8,401 proteins across all time points in the first donor sample (S5C Fig). Two comparisons of most physiological relevance were comparison of day 12 (termed "late exhaustion") versus either control (termed "resting") or day 6 cultures (termed "pre-exhaustion"). For these comparisons, 6,094 and 6,122 proteins, respectively, were deemed to be significantly differentially expressed (adjusted $p$-value <0.05) (Fig 2A). Among the proteins identified, many were exhaustion-associated (PDCD1, LAG3, ENTPD1, TOX, etc.), important CD4+ T-cell factors (CD40L, EOMES, GNG4, etc.), or T-cell signaling modulatory proteins (SLAM proteins, CD200, ICOS, CD27, CD28, etc.).

As an initial assessment of reliability, we examined our differential expression data for known markers and drivers of exhaustion (Fig 2B). Specifically, we detected significant changes in the expression levels of the highly characterized inhibitory receptors PD-1, LAG-3, TIM-3, and TIGIT, that are known to be concurrently expressed on exhausted T cells [62], as well as multiple other previously reported markers. GNG4, recently identified to demarcate exhausted CD4+ T-cell population [63], was also significantly up-regulated in the time course (Fig 2B).

Expression profiles of the 4 major exhaustion-associated inhibitory receptors were significantly up-regulated across the time course (FDR <0.05, fold-change ranging from 3.16 to 17.8) by late exhaustion relative to resting (S1 Data and S4 Data), thereby validating the model. Likewise, virtually all additional markers such as CD244 (2B4/SLAMF4), ENTPD1 (CD39), CD38, SLAMF7, SIGLEC10, and BTLA were significantly up-regulated by late exhaustion, while CTLA-4 and CD44, however, were not significantly changed. While also associated with T-cell activation, the persistent elevated co-expression of factors such as PD-1, ENTPD1, and CD38, is strongly linked to immune exhaustion [62].

Likewise, we observed significant changes in the abundance of previously described transcriptional regulators of T-cell exhaustion, including TOX, BLIMP-1 (PRDM1), EOMES, BATF, NR4A1/3, IRF4, ID2, and NFAT-family proteins. Notably, the transcription factor TOX, which is required for entry of CD8+ T cells into the exhausted state [23,64,65], showed progressive up-regulation in each time point relative to resting cell cultures. Additional evidence of exhaustion arises from our observation that Interleukin-7 Receptor Subunit Alpha (IL7R) protein expression is down-regulated in our chronically stimulated cultures. Specifically, this receptor is highly expressed in memory cells, as we observed in our resting condition, and is down-regulated when cells become exhausted [24]. We also observed an increase in both granzyme B and perforin at day 12, showing similar slight gain of function as seen in terminally exhausted CD4+ [51,52] and CD8+ T cells [53].

We compared our data set to exhaustion transcriptomic (mRNA) profiles reported in other studies. While CD4+ T-cell exhaustion data sets are largely lacking in the public domain,

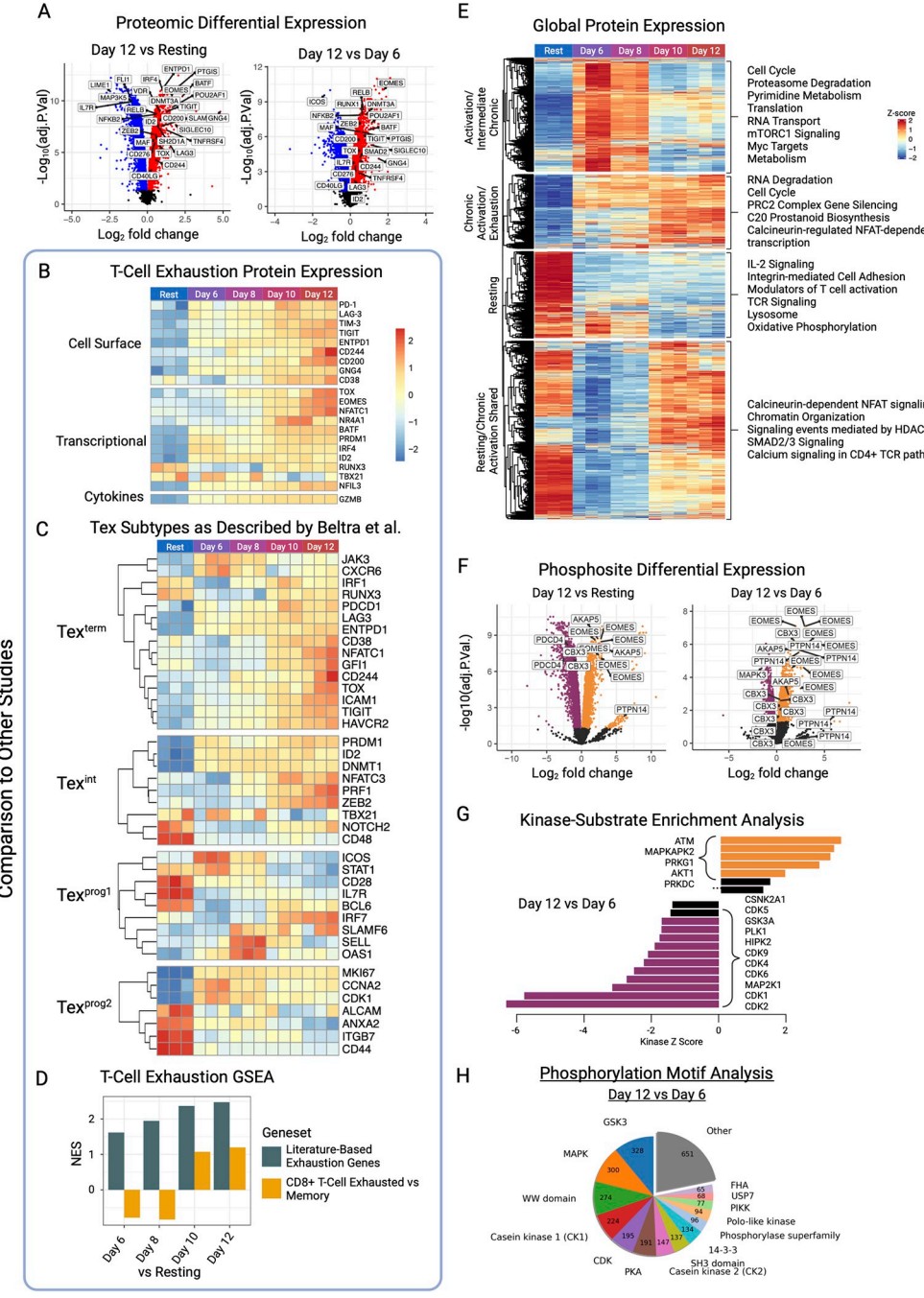

**Fig 2. Global proteomic and phosphoproteomic profiling reveals signaling dynamics while recapitulating exhaustion markers.** (A) Differential protein expression in late exhaustion (day 12) versus resting, and late exhaustion (day 12) versus early exhaustion (day 6) populations. (B) Expression patterns of select T-cell exhaustion-associated proteins. Rows z-score normalized. (C) Expression patterns of putative exhaustion signature markers defined by Beltra and colleagues [60]. Rows z-score normalized. (D) Enrichment analysis of proteomics data using an exhaustion-associated geneset and a CD8+ T-cell exhaustion versus memory geneset derived from differential expression of RNA sequencing data (Wherry and colleagues [61]). (E) Global proteomic expression grouped according to expression profile by hierarchical clustering. Genes from each cluster were analyzed by Enrichr for pathway enrichment results. Rows z-score normalized. (F) Differential patterns of phosphosite abundance in late exhaustion (day 12) versus resting cells, and late exhaustion (day 12) versus early exhaustion (day 6). (G) Kinase-substrate enrichment analysis of differential phosphosites between late (day 12) and early (day 6) exhaustion. (H) Phosphorylation motif analysis showing predicted kinase classes implicated for observed changes in phosphorylation between late (day 12) and early (day 6) exhaustion. The data underlying this figure can be found in either S1 Data under proteomics or under phosphoproteomics, and D and G can be directly found in S3 Data. Figure created with BioRender.com.

Beltra and colleagues used flow cytometry and RNA-seq to identify 4 subsets of CD8+ T-cell exhaustion states termed "*terminal*," "*intermediate*," "*progenitor 1*," and "*progenitor 2*" [60]. Comparing our protein expression data to these gene clusters, we observed that day 12 chronically stimulated CD4+ cells closely resemble the "*terminal*" exhausted CD8+ cells (**Fig 2C**). We also observe some agreement with the "*intermediate*" phenotype, which is likely explained by heterogeneity innate in bulk cell population analysis.

Next, we employed Geneset Enrichment Analysis (GSEA) to further compare our proteomic results to previously reported exhaustion gene profiles. Using 2 genesets, one containing curated exhaustion proteins from the literature and the other from RNA-seq of differential gene expression measured in exhausted versus memory CD8+ T cells in mice [61]. We observe that over time during chronic stimulation, the normalized enrichment score (NES) of these signatures within our proteomic data set increases and reaches its peak by the final day 12 time point (**Fig 2D**).

Recapitulation of the dynamics of known exhaustion-associated proteins provided confidence in the model system. To verify the consistency of our findings, we nevertheless evaluated the experimental results obtained from the replicate second donor, again evaluating the surface expression of well-defined exhaustion markers, cytokine phenotyping, and unbiased quantitative proteomics for global protein expression. Notably, we observed high overall agreement in the patterns of inhibitory receptor expression based on flow cytometry and proteome expression (**S1 Fig**), pointing to the overarching robustness of our main findings.

## Up-regulation of PRC2-mediated gene repression, prostanoid metabolism, and other select pathways in exhausted cells

We next examined proteome dynamics at a global level to look for trends across pathways and biological functions. Hierarchical clustering of the global set of proteomic profiles pointed to 4 major clusters of protein expression throughout the progression to exhaustion (**Fig 2E**). These patterns were defined by: (1) "*Activation/Intermediate Chronic*," wherein protein expression was elevated in early chronic stimulation time points (days 6 to 8) but down by late chronic stimulation/exhaustion as well as resting cultures; (2) "*Chronic Activation/Exhaustion*," which corresponds to elevated expression in late chronic stimulation time points (days 10 to 12) relative to other time points, corresponding to more terminal/late exhaustion; (3) "*Resting*," where protein expression is highest in resting cells, and then decreases by day 6, remaining low through day 12; and (4) "*Resting/Chronic Activation Shared*," which shows an inverse pattern of the first cluster, with decreased protein expression at intermediate time points relative to resting and late chronic stimulation/exhaustion.

The "*Activation/Intermediate Chronic*" cluster was predictably enriched for biological processes and pathways linked to T-cell activation (**Fig 2E**), including cell cycle control, pyrimidine metabolism, translation, and mRNA transport that activated T cells deploy when undergoing proliferation following initial antigen exposure. The mTORC pathway was also active during this stage, promoting cell proliferation, survival, growth, and migration. mTORC, along with Myc, also controls cellular metabolism, prompting T cells to ramp up energy production via glycolysis, a critical early step during T-cell activation. Interestingly, we observed proteosome-dependent degradation, which has been linked to regulating T-cell metabolism and fate specification in CD8+ T cells [66].

In the "*Chronic Activation/Exhaustion*" cluster, we observed enrichment of cell cycle factors, in this case reflecting the marked down-regulation of proliferation in this cell population (for a comparison of the different proteins related to cell cycle that appear in "*Chronic Activation/Exhaustion*" and "*Activation/Intermediate Chronic*" clusters, see **S1 Data**). We also

observed enrichment of genes involved in RNA degradation, which could be explained by RNA turnover being used as a means for turning off gene expression in cells undergoing exhaustion [67]. We also found supporting evidence that NFAT-mediated target gene expression is occurring, in agreement with the up-regulation of NFAT-related proteins in the proteome (NFATC1, NFATC2, NFATC3, NFAT5). Interestingly, chromatin remodeling PRC2 Complex, which mediates long-term gene silencing through modification of histone tails, was elevated in day 12 cultures, suggesting a regulatory role in controlling gene suppression during chronic stimulation. We also observed pathways related to cell death, which may drive a portion of the cells to undergo apoptosis under exhaustion-inducing conditions. Lastly, we found evidence of metabolic change, most intriguingly pointing to C20 Prostanoid Biosynthesis. Specifically, the C20 class of prostanoids are comprised of prostaglandins, thromboxanes, and prostacyclins, which have been implicated in the regulation of the immune response in T cells [68,69].

The "*Resting*" cluster was enriched for common T-cell processes. Some were, however, related to T cell activation, which may be caused by the fact these proteins can be down-regulated by the chronic time points to negatively control activation. These pathways include IL-2 signaling, cell–cell adhesion pathways, modulators of T-cell activation and TCR signaling, and some metabolic functions such as oxidative phosphorylation, a resting T cells primary energy source. Example proteins from these pathways that are up in resting and go down with chronic activation include IL2RB, CD3E and CD3G, LCK, CD4, and many OXPHOS proteins such as those in the ATP synthase complex like ATP5B/E/I. Notably, this observation suggests however that exhausted T cells are not using oxidative phosphorylation to the same extent that resting cells are, which is consistent with the findings that impaired mitochondrial function and OXPHOS is associated with T-cell exhaustion [70].

The last cluster, "*Resting/Chronic Activation Shared*," represents proteins that are up-regulated in both resting and late chronic stimulation relative to the earlier stimulation time points and has the largest number of components. Proteins in this grouping are involved in cell maintenance and suppression of activation, proliferation, migration/cell–cell interactions, and other biological processes that are shared by the 2 phenotypes. Furthermore, we also found enrichment of signaling factors linked to CD4+ TCR pathway, calcineurin-dependent NFAT signaling, epigenetic chromatin organization, and HDAC-mediated signaling, further pointing to the importance of coordination of metabolic changes with epigenetic and transcriptional control through signaling cascades. We should note that evidence of calcineurin-dependent NFAT signaling pathway was found in both "*Resting/Chronic Activation Shared*" (CREBBP, MAP3K1, YWHAB, PRKCB, BAD, PRKCE, NFATC3, NFATC2, PRKCA, NFATC1, AKAP5, CABIN1, FKBP1A, PPP3CA, MAPK9, PPP3CB, CAMK4, CHP1, EP300, MEF2D, MAPK3) and "*Chronic Activation/Exhaustion*" (ITCH, ERG3, MAF, IFNG, IRF4, CASP3, IL2RA, SLC3A2, CBLB, PTGS2, TNF, JUNB). While the proteins contributing to the signals differed in each cluster, a majority of these proteins (from both clusters) are up by day 12, suggesting that day 12 is where the strongest signal comes from.

## Post-translational regulation of exhaustion factors and signaling pathways

To directly assess changes in signaling pathways arising from chronic stimulation, we performed phosphoproteomic analysis of the same samples to capture global protein phosphorylation changes during the progression of exhaustion. After stringent filtering, we quantified 11,728 unique high-confidence phosphosites on 2,685 phosphoproteins. A total of 7,499 phosphorylation events (on 2,100 proteins) were found to be significantly differentially regulated (FDR <0.05) between pre-exhaustion (day 6) and resting, 6,618 phosphosites (on 1,914

proteins) between late exhaustion (day 12) and resting, and 3,166 (on 1,158 proteins) between late exhaustion (day 12) and pre-exhaustion (day 6) (**Fig 2F**).

Reflecting the dramatic impact of TCR-mediated signaling events, most changes occurred between resting and day 6 cells, including phosphorylation of many TCR pathway components, cell cycles, and cytoskeletal proteins. We observed less significant phosphorylation changes between day 6 and 8, but increased phosphorylation of exhaustion- and CD4+ memory cell-associated transcription factor EOMES, as well as decreased phosphorylation of MAP kinases. Significant phosphorylation changes were observed between days 8 and 10 where increased EOMES phosphorylation was sustained. Changes in phosphorylation were minimal by day 12 compared to day 10, with only tens of phosphosites significantly changing.

Other notable T-cell regulatory proteins include NFATs, known to exert key transcriptional roles in T cells, which were highly phosphorylated in our data set, indicating that many sites significantly changed over time. For example, we detected and quantified 17 phosphosites on NFAT1 (NFATC2), some of which were significantly up-regulated in late exhaustion (day 12) (S99, S110, S243, S330, S757/759, T224) or pre-exhaustion (day 6) (S230, S236, S243, S757/759, S814, and T224). We also picked up 9 phosphosites on NFAT2 (NFATC1), some that have been functionally annotated and were significantly differentially regulated. Serine 245, significantly hyperphosphorylated in late exhaustion compared to pre-exhaustion, is a known substrate of kinases PKACA and PIM1, impacting intracellular localization, transcriptional activity, and cell motility. We also quantified 9 unique phosphosites on NFAT4 (NFATC3), of which S163, S165, and S207 were detected and have annotated function. However, these phosphosites were not significantly different between time points, except for S163, which is dephosphorylated in day 8 and day 10 versus resting, and S207, which was dephosphorylated on day 6 and 8 compared to resting. S163 is known to be phosphorylated and inhibited by JNK2 [71], affecting localization and transcription. S207 is known to be phosphorylated by CK1A, which also controls subcellular localization. Such observed phosphorylation patterns suggest that NFAT4 is cytoplasmic during the resting cell state but enters the nucleus during the early stages of chronic activation (day 6/8) before returning to the cytoplasm during the terminal exhaustion state (day 10/12).

We observed increased phosphorylation on proteins such as EOMES, PTPN14 (PEZ), NAB2, and AKAP5 in late exhaustion (day 12) compared to resting (**Fig 2F**). EOMES was hyperphosphorylated at many sites, especially in the T-box associated domain between serine 630 and serine 655. AKAP5, which interacts with important T-cell signaling proteins such as PKA, PKC, and calcineurin and inhibits IL-2 transcription by disrupting calcineurin-dependent dephosphorylation of NFAT [72,73], was observed to have 2 sites that were highly phosphorylated over time with chronic stimulation. Chronic antigen exposure induced pronounced phosphorylation of the phosphatase PEZ (**Fig 2F**), encoded by PTPN14, and a major regulator of adheren junctions and cell–cell adhesion. NAB2 is also seen highly phosphorylated, which is known to enhance T cell function. As increased phosphorylation is observed in parallel with loss of effector function, it is possible this protein is inhibited by phosphorylation of these sites.

A subset of phosphosites with increased fold change and significance appears slightly separated from the main cluster (**Fig 2F**). Notably, these phosphosites tend to follow the same phosphorylation dynamics of up-regulation in all stimulation time points compared to resting but staying consistent from day 6 until 12. Corresponding phosphosites were on proteins enriched for pathways such as cell proliferation (G2/M phase control, mitotic spindle formation) and T-cell maturation (Enrichr, q value < 0.1).

## Global phosphosite trends reveal pathway dynamics during chronic stimulation

To glean insights into signaling kinetics, we clustered each phosphosite profile, which revealed 4 basic expression patterns. The first cluster showed a phosphorylation pattern that was low in resting cells but stably hyperphosphorylated from pre- to late-exhaustion. This cluster corresponded to regulators of the cell cycle, transcriptional repression via RanBP2, control of gene expression by vitamin D receptor (VDR), histone methylation (epigenetic control), and YY1 activity, an important regulator of exhaustion [56] (**S6D Fig**). VDR, a nuclear receptor that is known to play an important role in T-cell function, regulates hundreds of genes [74]. Interestingly, we observed up-regulation of VDR upon T-cell activation by day 6, but only a slight increase by day 12 of chronic activation. Upon binding the active form of its ligand D3, VDR translocates into the nucleus of T cells and binds to promotor regions of PDCD1 (PD-1), TIM3, and TIGIT, inhibiting their expression, thereby decreasing the exhaustion phenotype [75]. Hence, the up-regulation we observed could act as an "exhaustion modifier," supported by studies showing reduced disease severity and T-cell exhaustion in SARS-CoV-2 patients with higher levels of vitamin D [76,77].

In the next cluster, phosphorylation peaks by day 6 to 8 and becomes dephosphorylated again by late exhaustion. This cluster corresponded to various biological responses expected to emerge from activated T cells such as cell cycle control, mRNA processing, mTOR signaling, pyrimidine biosynthesis and salvage, PLK1 signaling, and Aurora B signaling.

The subsequent cluster represents proteins heavily phosphorylated in resting cells, which were dephosphorylated by day 6, and generally stayed at low levels through to late exhaustion. These factors corresponded to IL-2 signaling (FYN, GAB2, CBL, etc.), the TCR pathway (LAT, FYN, MAPK1, SOS1, RAF1), PI3K/mTOR/AKT signaling (AKT1S1, CAMK4, MAPK1, MYC, etc.), protein secretion (GBF1, GNAS, GOLGA4, etc.), and Myc targets (SSB, TRA2B, TRIM28, U2AF1, etc.), all of which are known to be down-regulated during chronic stimulation and exhaustion [10].

Similar to the cluster in our proteomics data set, the last cluster generally followed a pattern of dephosphorylation by day 6, gradually restoring phosphorylation to resting levels by day 12. These components were enriched for chromatin remodeling enzymes, as well as components of T-cell activation and mRNA processing.

## Differential kinase activity in chronically stimulated CD4+ T cells

To determine which protein kinases exerted the largest impact in driving these signaling events and, by inference, exhaustion progression, we leveraged knowledge of annotated substrates and phosphorylation motif information to infer specific kinase or kinase-family activity in our model.

First, we performed kinase-substrate enrichment analysis on our phosphosite data (**Fig 2G**). Relative to pre-exhaustion (day 6), we observed an increase in ATM, MAPKAPK2 (MK2), PRKG1, AKT1, and PRKDC kinase activity by late exhaustion (day 12), and a sharp decrease in CDK2, CDK1, MAP2K1, CDK6, CDK4, CDK9, HIPK2, PLK1, GSK3A, CDK5, and CSNK2A1 kinase activity (**Fig 2G**). Inhibition of the top hit, ATM, has already been shown to restore exhausted CD8+ T-cell function in an HCV-infection context [78]. Together with p38, MAPKAPK2 (MK2) signals in response to cell stress or TCR/CD28 engagement [79]; chronically stimulated T-cells experience both. AKT1 has many roles in T-cell function and development, but in late-stage differentiated T cells, inhibition of AKT1 has been shown to enhance tumor elimination [80,81]. Up-regulation of PRKDC, linked to DNA damage repair, may reflect increased cell stress experience during T-cell exhaustion. Conversely,

kinases prominently down-regulated in terminally exhausted (day 12) cells were cell cycle-dependent kinases, consistent with concomitant decrease in cell proliferation.

Additionally, we used phosphorylation motifs that were differentially phosphorylated to predict kinase-class participation of these changes (**Fig 2H**). When we compared late exhaustion to pre-exhaustion, we observed a large percentage of phosphorylation events were caused by kinases GSK3, MAPK, WW domain CK1, CDK, and PKA, among others, providing further insight into kinase activity that takes places between pre- and late-exhaustion.

## Expression trajectories implicate CD276, FLT-1, and SLC proteins in CD4 + T-cell exhaustion

To hone in on individual proteins (versus pathway level) that are likely contributing to the exhausted phenotype, we devised a partitioning strategy for sifting through the data with the aim of extracting physiologically relevant groupings. Since proteins with similar expression patterns often play roles in similar processes [82], we examined profile similarity to observe coherent patterns during progression of exhaustion. We first divided proteins by functional class implicated in T-cell exhaustion, specifically cell surface proteins and transcription factors. Next, we applied k-means clustering to group each type according to their expression patterns to identify clusters enriched for exhaustion-associated proteins to implicate other potential regulators exhibiting similar profiles. Notably, the resulting tight groupings revealed sets of highly correlated cell surface proteins and transcription factors that show low expression in resting cells and increasingly elevated expression throughout the progression of exhausted phenotypes reaching peak expression at day 12 (**Fig 3A and 3B**).

Strikingly, clusters that were characterized by low expression in resting cells and increasing expression throughout progression of the exhaustion phenotype (**Fig 3A and 3B**) were significantly enriched for many known exhaustion-associated factors including inhibitory receptors PD-1, TIM-3, LAG-3, TIGIT, and ENTPD1, and exhaustion-associated transcription factors TOX, BLIMP-1, BATF, and NFATC1. Such observations implicate candidate novel associations not previously linked to T-cell exhaustion that may play a role as drivers or markers of CD4+ T-cell exhaustion based on the coherence of their expression profiles. We also observed a cluster that follows a similar profile in our kinase clustering, which contains a variety of kinases from different classes, including MAP kinases, CDKs, NEKs, and STKs, among others (**S4 Data**). When we performed an integrative analysis (using eXpression2Kinases [83]) to investigate whether these kinases or transcription factors may be regulating each other, we found that 2 transcription factors from the clusters shown in Fig 3B were enriched for targets in our kinase cluster (cluster 3 in **S4 Data**) that follows the same expression pattern; NFYB targeted 11 out of 21 kinases from this list, and TCF3 targeted 5 out of 21 kinases (**S5 Data**), suggesting these transcription factors may be playing a role in the regulation of kinases from this cluster.

Besides the most widely known receptors (PD-1, LAG-3, TIM-3, TIGIT), the cluster of cell surface proteins (84 genes total) also contained a few recently characterized inhibitory receptors (CD244/SLAMF4, SLAMF7, SIGLEC10), as well as many novel proteins that have not previously been associated with chronic stimulation/exhaustion (**S6 Data**). CD244, also known as SLAMF4, is a member of the signaling lymphocytic activation molecule (SLAM) family and has been shown to have both stimulatory and inhibitory activities [84,85]. However, in the context of chronic infections, it has been suggested that CD244 limits CD8+ effector responses [86]. Furthermore, this gene has also been previously associated with exhaustion in patients with chronic lymphocytic leukemia as a marker of T-cell exhaustion. However, cells displaying CD244 still produced some cytokine while still losing cytotoxicity and proliferation capacity

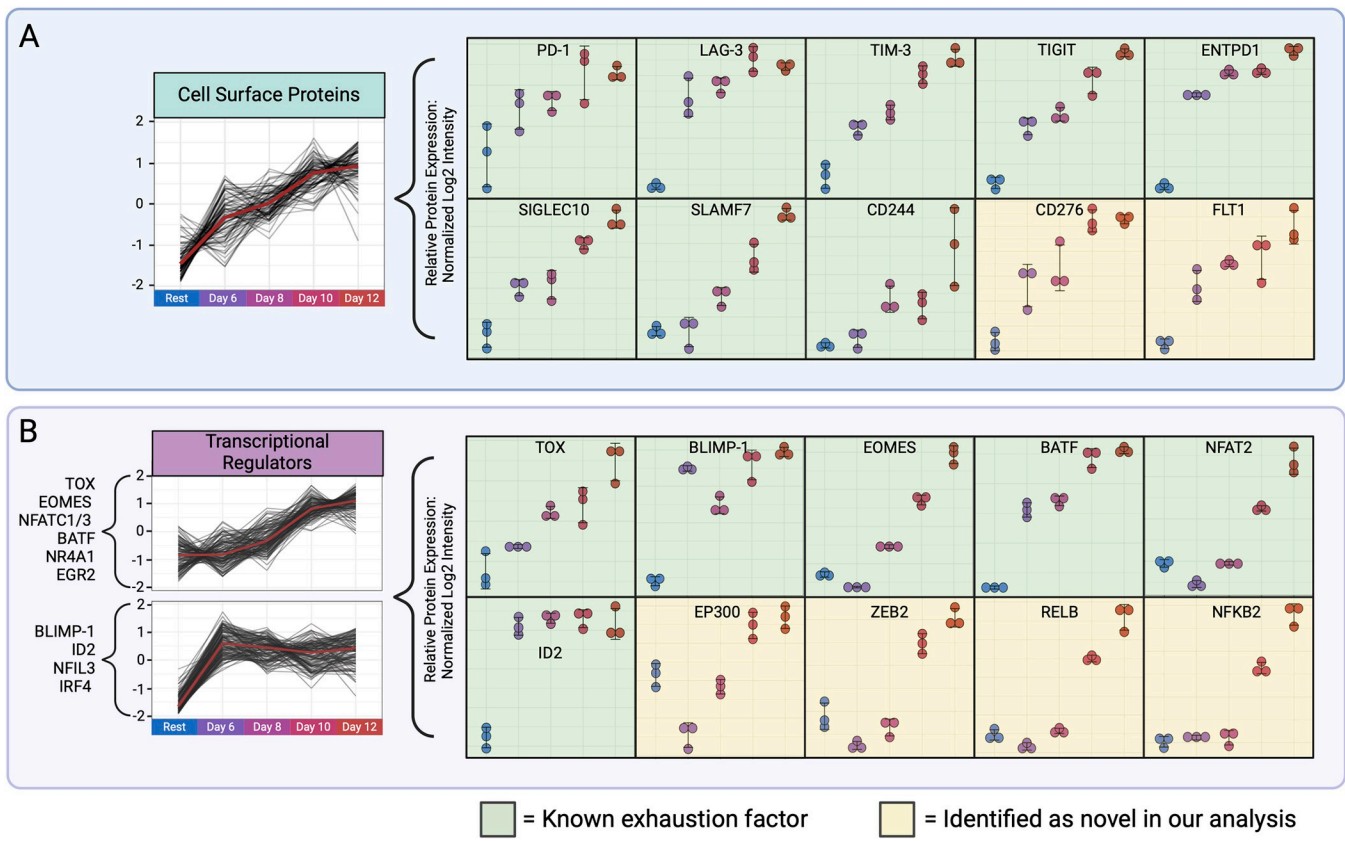

**Fig 3. Hierarchical clustering of cell surface and transcriptional regulator protein expression reveals clusters enriched for known exhaustion-associated proteins.** (A) Cell surface protein cluster (left) enriched for exhaustion-associated proteins, with known (green) and novel (yellow) protein profiles highlighted (right). (B) Transcriptional regulator protein clusters (left) enriched for exhaustion-associated proteins, with known (green) and novel (yellow) protein profiles displayed (right). The data underlying this figure can be found in S1 Data. Figure created with BioRender.com.

[87]. The self-ligand SLAMF7 was recently implicated as a player in T-cell exhaustion, and its signaling was shown to phosphorylate STAT1/3, expression of multiple inhibitory receptors, and expression of transcription factors associated with T-cell exhaustion [88]. SLAMF7 signaling activity is mainly controlled by the presence/absence of cytoplasmic adaptor proteins SH2D1A/B, which follows the same expression pattern in our data as the receptor itself (SHD1A detected, significantly up-regulated by day 12, FDR <0.05). SIGLEC10, a sialic acid-binding immunoglobulin-like lectin, has also been implicated in having immunosuppressive functions [89,90] and was identified as being up-regulated on activated CD4+ T cells [91], but has not yet been directly linked to exhaustion. In our data set, this gene is not only up-regulated on pre-exhausted cells (day 6), but even more highly expressed by late exhaustion (day 12), suggesting that SIGLEC10 is involved in the exhaustion phenotype of these CD4+ T cells.

Among the proteins in the cell surface cluster were a few receptors/cell surface proteins that have been linked to dampening of T-cell responses in various aspects but have not yet been linked to exhaustion. CD276 (B7-H3) is a member of the B7 superfamily, which is a group of receptors that play modulatory roles during T-cell responses. This molecule, originally suggested to have both costimulatory and inhibitory effects in murine studies, was shown to be inhibitory in human T cells [92]. Although shown to down-regulate T-cell activation, proliferation, and cytokine production, CD276 has not yet been linked to chronic T-cell stimulation or exhaustion. We observed significant up-regulation over time, with resting cells expressing

the least amount of CD276, pre-exhausted cells (day 6) expressing an intermediate amount, and exhausted cells (day 12) expressing the most, suggesting that this molecule may be up-regulated by CD4+ T cells in response to repeated antigen stimulation to dampen T-cell responses.

Another protein of interest in this cluster is Fms Related Receptor Tyrosine Kinase 1 (FLT-1). This receptor is a part of the vascular endothelial growth factor receptor (VEGFR) family, specifically bound and activated by VEGF-A, which is often secreted by tumors into the micro-environment and drives TOX-dependent T-cell exhaustion [93]. Since we observed up-regulation over time in exhausted CD4+ T cells following the same expression pattern as other inhibitory receptors associated with exhaustion, up-regulation of FLT-1 may contribute to this cell state.

## Independent marker validation of in vivo phenotypes using human clinical samples

To investigate if the exhaustion patterns and markers seen in vitro also occur in vivo, we first analyzed the surface expression of CD276 and FLT-1 on T cells present in peripheral blood. Five HIV+ and 5 HIV- donor peripheral blood mononuclear cell (PBMC) samples were analyzed by spectral flow cytometry using validated antibodies specific for these factors (**Figs 4 and S7**). While expression of these 2 markers on CD4+ T cells in both HIV–positive and HIV–negative donors was low overall, we found that CD4+ T cells positive for either marker (CD276 or FLT-1) showed concomitantly significantly higher PD-1 expression, establishing both that these markers do indeed appear on cells in vivo, and implying that they may mark functionally exhausted T cells. We found this trend to be true regardless of HIV status, and we did not see a significant difference between HIV–positive and HIV–negative donors. However, this may be due to the effectiveness of modern HIV treatments, as these patients showed similar T cell subset frequencies and protein expression as the HIV–negative donors.

Lastly, we observed an enrichment of solute carrier (SLC) family proteins (**S6 Data**), which mediate import and export of diverse small molecules (e.g., ions, peptides, nutrients) needed for metabolism. As T cells are heavily regulated by metabolic changes [11,12,94], these transporters may contribute to exhaustion via modulation of metabolism, again hinting at the major metabolic shifts immune cells undergo.

## Transcriptional regulators altered in chronically stimulated CD4+ memory T cells

Significant and long-lasting metabolic changes are often strengthened and controlled via transcription. Transcriptional control of exhaustion is also known to be critical and therefore the same clustering strategy as described for the cell surface proteins was next applied to transcriptional regulators to identify potential novel transcription factors or co-factors. We observed 4 major protein expression patterns (**S7 Data**). We again saw a promising cluster that displayed the pattern of low in resting and increasing during chronic stimulation (**Fig 3B**). Similar to observations of cell surface proteins clusters, this cluster was enriched for exhaustion-associated transcription factors (TOX, EOMES, NFACT1 and 3, BATF, NR4A1, EGR2). We also observed other exhaustion-associated factors in a cluster that follows a somewhat similar pattern, but expression comes up earlier and stays high, and the actual expression patterns of the factors found here show expression profiles that might fit into the previous cluster. The exhaustion-associated factors found here were BLIMP-1 (PRDM1), ID2, IRF4, and NFIL3 (**Fig 3B**). These results suggest that there may be novel proteins following similar expression profiles identified here.

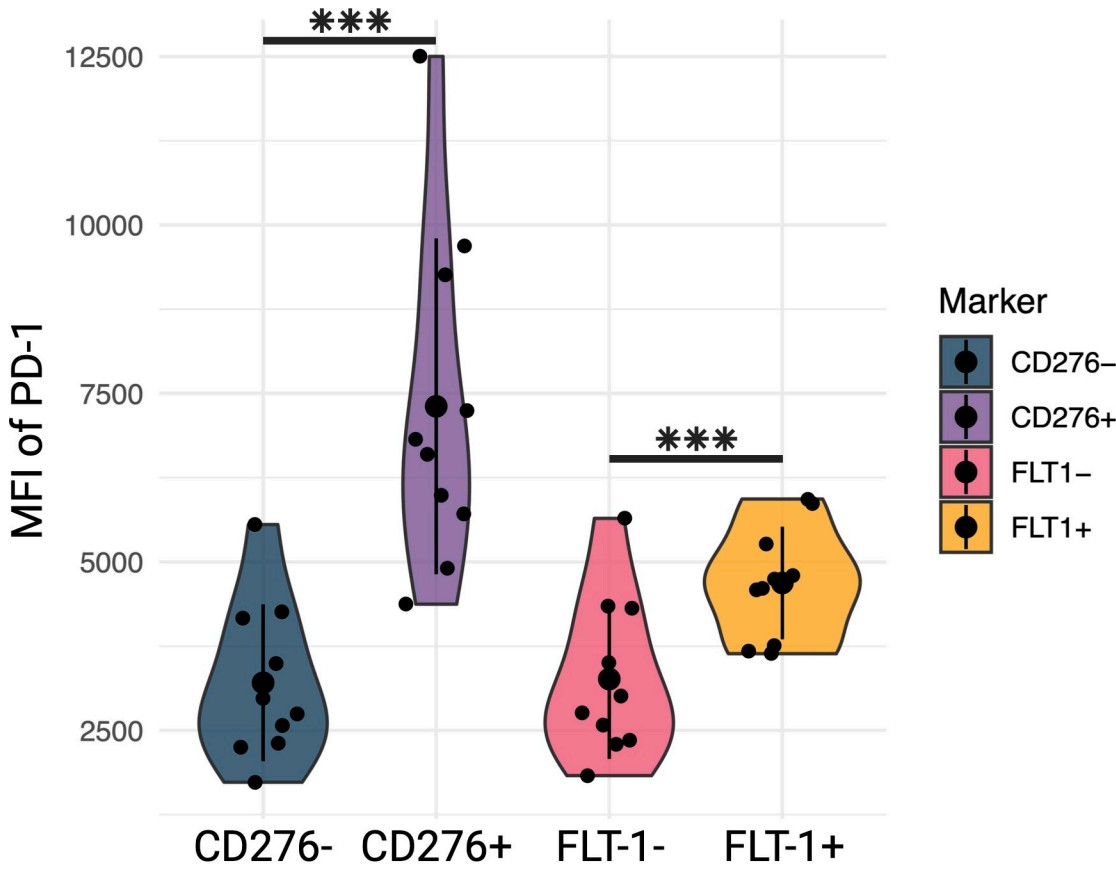

**Fig 4. Flow cytometric analysis of HIV+ and HIV- primary T cells reveals increased PD-1 expression in CD276+ and FLT-1 + CD4+ T cells.** Mean fluorescence intensity of PD-1 in CD4+ T cells from both HIV+ and HIV- patients (grouped due to observing same effect in both) for CD4+ cells that are CD276 or FLT-1+. The data underlying this figure can be found in the supplemental flow cytometry files uploaded to flowrepository.org. Figure created with BioRender.com.

One protein identified in this cluster was MAF BZIP Transcription Factor (c-Maf, MAF), a leucine zipper-containing transcription factor. While not as well studied as factors like TOX or BLIMP-1, the role of MAF has been investigated in CD8+ T cells, where its overexpression leads to repression of cytokines IL-2 and IFN-γ production, and an increased expression of T cell exhaustion genes, such as PD-1 [62,95]. In another study, MAF was suggested to even be a driver of exhaustion, where CD8+ T cells also had increased anti-tumor capacity when MAF was knocked out [96]. Lastly, MAF induced exhaustion-associated genes in mice, such as PD-1, TIGIT, TIM-3, and LAG-3, in cooperation with the known exhaustion transcription factor, BLIMP-1 (PRDM1) [97]. However, MAF is not up-regulated by day 12 in our second donor, conflicting with this evidence.

Another interesting transcription factor found in this cluster is transcription factor zinc finger E-box binding homeobox 2 (ZEB2), which is a member of the Zfh1 family of 2-handed zinc finger/homeodomain proteins. While its significance in CD4+ T cells has not been closely studied, ZEB2 is a transcriptional repressor and is involved in terminal differentiation of CD8 + T cells (effector and memory) [98,99]. It has been previously reported that ZEB2 is up-

regulated in murine CD4+ T cells by day 7 during an LCMV infection [100]. While ZEB2 appears to be an indicator of terminal effector-like cells, ZEB2 may also be playing a role in the CD4+ T cell chronic infection dysfunctional phenotype, directly repressing both IL7R and IL-2, which are 2 indicators of exhausted CD4+ T cell phenotype. ZEB2 has also been shown to correlate with the expression of ID2 and T-bet, suggesting their involvement in a larger transcriptional network [100]. Further, ZEB2 has also been found in a gene signature differentially implicated in exhaustion versus memory, which was also grouped alongside TOX and EOMES [101].

Besides individual factors of interest, we also observed enrichment of factors suggesting signaling pathways and complex up-regulation, for example, downstream components of non-canonical NF-κB signaling. The downstream transcriptional program of non-canonical NF-κB signaling is driven by a heterodimer of RELB and p52 (NFKB2), both of which are highly significant in the same cluster as transcription factors such as TOX, EOMES, and NFATC1. However, we did not find the same significant up-regulation of canonical NF-κB signaling factors p50 (NFKB1), REL, and RELA. In non-canonical NF-κB signaling, p100 is a pre-cursor to p52, and it is designated for processing into p52 through phosphorylation. We observed up-regulated phosphorylation of serine 707 of this protein at day 12, but decreased phosphorylation during days 6 to 8, which has been annotated as being phosphorylated by GSK3b and promotes it degradation by FBXW7α. Such an observation perhaps suggests that this pathway is active during early exhaustion and may be primed but inhibited through GSK3 activity in late exhaustion. Such characteristics could also be part of the mechanism of inhibition by GSK3b and its ability to somewhat improve functionality of exhausted cells when inhibited [102,103]. Target genes of non-canonical NF-κB pathway have been shown to include BLIMP-1 (via RelB, albeit in tumor cells) [104], suggesting this gene may be an early regulator of the exhaustion transcriptional network.

Another observation in this cluster was the enrichment of subunits of the polycomb repressive complex 2 (PRC2), providing further evidence from our pathway enrichment results where PRC2 complex repressive activity was first identified. We detected 8 out of 9 members of this complex (JARID2, AEBP2, EED, EZH1, EZH2, SUZ12, RBBP7, RBBP4), all of which were significantly up-regulated by late exhaustion (day 12) when compared to resting (**S7 Data**). The PRC2 complex is an epigenetic modifier, which functions by methylating H3K27 histones on target genes, usually suppressing expression. EZH2 and the PRC2 complex have been well studied and are known to play a role in development, proliferation, and function of T cells, as well as potential immunomodulators [105]. Recently, it was found that PD-L1 expression on exhausted CD4+ T cells was regulated by PRC2 and SWI/SNF [106]. While the opposite was observed in tumor cells, with PRC2 up-regulation leading to down-regulation of PD-L1, it is interesting to see that anti-CLTA-4 immune checkpoint inhibitor therapy is actually improved when given in conjunction with EZH2 therapies [107].

## Network model of dysregulation in exhausted CD4+ memory T cells identifies p300 as a key node driving phenotype

To identify key drivers of the chronically stimulated dysfunctional phenotype we defined molecularly, we incorporated both our proteomic and phosphoproteomic data sets by using a causal network analysis across time that allows us to predict proteins in our model that are responsible for large changes of expression or phosphorylation.

Using known and annotated interactions such as protein–protein interactions or phosphorylation events, we examined the networks of differential phospho/proteins in each time point which factored in the previous comparison starting from resting cells. We solved a Steiner tree

problem, connecting the largest number of differential features (leaves) to upstream regulators (connectors). In our final network (**Fig 5A**), which represents late exhaustion (day 12) of chronic stimulation, we observed several clustered subnetworks, with the largest revolving around p300 (EP300) in the top left corner (**Fig 5B**) with connections to a few important players in the T-cell exhaustion phenotype, including transcription factors EOMES, BATF, TBX21, and the cytokine IFN-γ. Furthermore, p300 is connected to PTPN2, a phosphatase known to inhibit T-cell activation by dephosphorylating key signaling proteins such as LCK, further suggesting p300 may play a role in T-cell function. In our proteomic data set, p300 is also significantly increased over time, peaking at day 12 chronic stimulation (**Fig 5B**).

To test if p300 is in fact playing a role in the phenotype, we inhibited p300 using A-485, a small molecule inhibitor. Specifically, cells were chronically stimulated for 10 days, and then treated with drug or vehicle for 2 days alongside the final 2 days of stimulation (**Fig 5C**), and then analyzed cells for basic exhaustion-associated phenotype using flow cytometry. Cells treated with p300 inhibitor showed significantly reduced expression levels of inhibitory receptor and exhaustion markers PD-1 and TIM-3 (**Fig 5D and 5E**). While cells still had expression of these receptors, cells treated with p300 inhibitor overall had on average significantly lower expression, and increased number of cells with low expression (**Fig 5D**). To better quantify this change, we looked at mean fluorescent intensity (MFI) values of drug treated versus DMSO treatment and observed significant decrease in receptor expression (**Fig 5E**). As 2 controls, we treated cells with Dasatinib to block T-cell activation or removed beads from day 10 until day 12 (rested for 2 days). Since p300 is known to play a role in cell cycle, we wanted to ascertain if the effects seen with p300 inhibition were not solely a consequence of blocking T-cell activation. If this was the case, we would expect PD-1/TIM-3 expression of p300-inhibitor treated cells to be equivalent to either resting or treating with Dasatinib, which was not the case. Further evidence that p300 inhibition was not blocking T-cell activation was that activation marker CD137 was unchanged in p300 inhibitor treated cells versus DMSO but was decreased in rested and Dasatinib conditions (**S8 Fig**).

P300 is a histone acetyltransferase cofactor, which imparts its effects via histone remodeling, usually activating bound genes. In our p300 subnetwork, we also observed that this gene has connections to HDAC2, which is a class I HDAC, along with HDAC1 and 3, suggesting further epigenetic regulation. T-cell fate and function is also known to be strongly controlled by epigenetic regulation, prompting us to investigate which genes p300 and other similar epigenetic-modifying cofactors are regulating.

## Control of transcriptional changes influenced by epigenetic co-factor binding dynamics

Epigenetic control is mediated by complexes of regulatory cofactors and transcription factors. Our analyses have identified roles for epigenetic cofactors (e.g., PRC2, p300, HDACs) and TFs, therefore, we sought to determine whether TF-cofactor interactions might further indicate key regulatory complexes driving T-cell exhaustion. To assess recruitment of epigenetic modifiers by sequence-specific DNA-binding factors, we performed biochemical assays using the CASCADE (Comprehensive ASsessment of Complex Assembly at DNA Elements) microarray platform [108]. In CASCADE, recruitment of cofactors by TFs is assayed by profiling the recruitment of cofactors to double-stranded DNA microarrays comprising hundreds of annotated TF-binding site sequences. We applied nuclear extracts from acutely stimulated (2 days) and 12-day chronically stimulated cells to the microarrays and measured cofactor recruitment using fluorescently labeled antibodies. Acutely stimulated cells were used here as a control to allow us to see how complexes form during healthy T-cell challenge compared to that of

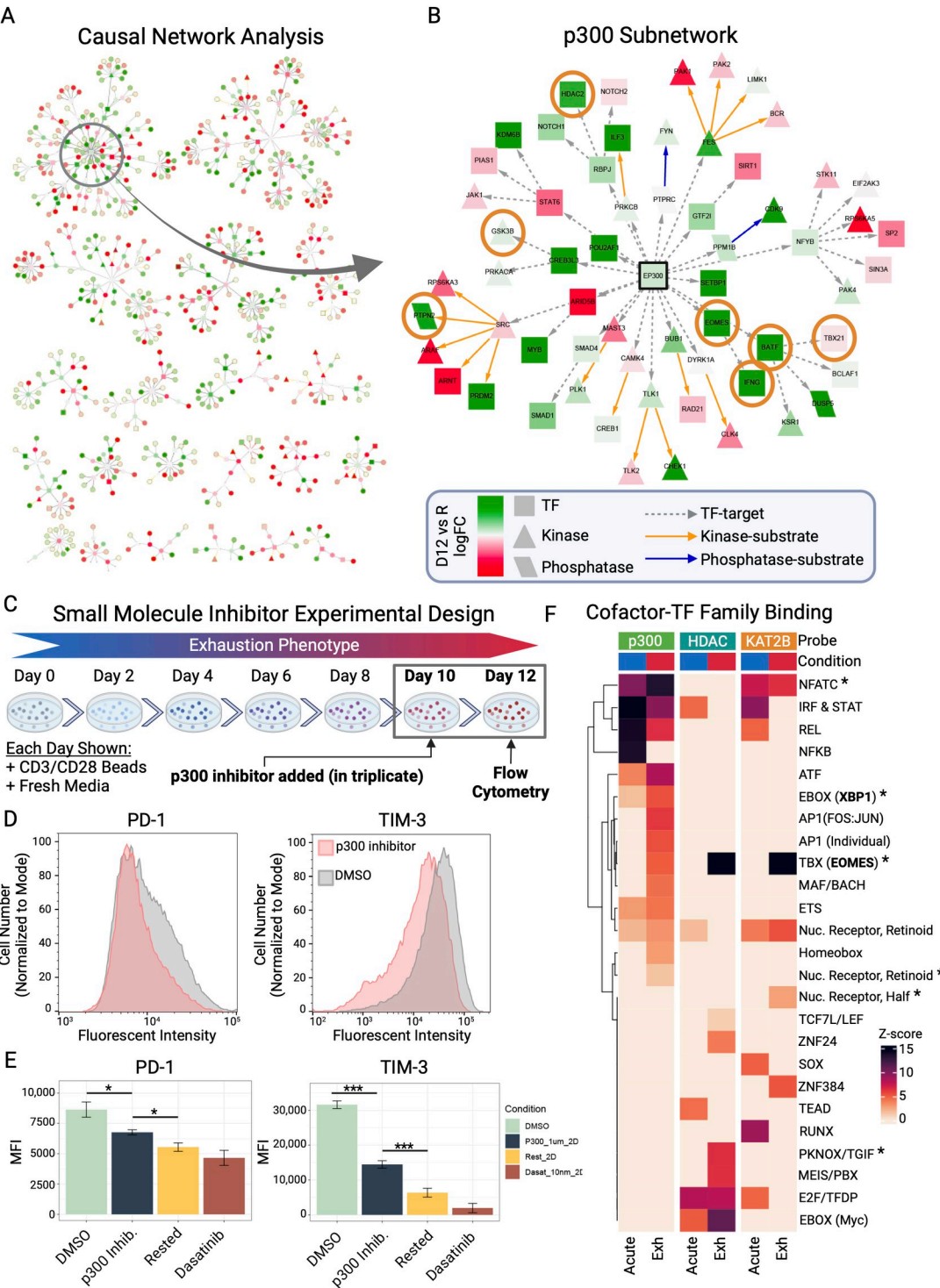

**Fig 5. Causal network analysis predicts p300 as regulator of exhaustion phenotype and p300 inhibition leads to down-regulation of inhibitory receptors PD-1 and TIM-3.** (A) Causal network analysis leveraging annotated interactions (transcription factor-target, kinase-substrate, phosphatase-substrate) as a framework to evaluate protein expression changes (late exhaustion (day 12) versus resting, taking into account previous time point comparisons) to predict key nodes (regulators) impacting most changes seen across the data set. (B) Pruned version of a large subnetwork with p300 (black outline) at main hub. Orange circles mark proteins of interest mentioned in main text. (C) Depiction of experimental design of p300 inhibition studies. (D) Flow cytometric analysis of PD-1 and TIM-3 showing shifts in fluorescent intensity of cell populations in treated

versus untreated (cell number normalized to mode). Cell numbers shown for each plot are 20,312 for p300 treated and 30,438 for DMSO, which corresponds to all CD4+ T cells from the sample (excluding doublets and debris, see Methods for gating), corresponding to >95% of sample. (E) Bar chart of mean fluorescent intensity showing significant changes (*T* test) in PD-1 and TIM-3 during p300 inhibition versus controls (antigen removed for 2 days, or treatment with T-cell activation inhibitor Dasatinib); *, *p*-value <0.05, ***, *p*-value <0.005. (F) Heatmap showing recruitment of cofactors (p300, class I HDACs, KAT2B) to transcription factor family DNA motifs as measured using CASCADE protein-binding microarrays. The data underlying this figure can be found in S8 Data and S9 Data (A, B), supplemental flow cytometry files uploaded to flowrepositroy.org (D), and the S3 Data (E, F). Figure created with BioRender.com.

exhausted cell challenge. We determined multiple cofactor recruitment motifs for each cofactor, which we used to infer the TF family to which a given cofactor was recruited by searching against TF motif databases (see **Methods**). Alongside p300, we opted to probe both KAT2B, a histone acetyltransferase known to associate with p300 that has been implicated as a marker in T-cell exhaustion [109,110], and a mixture of class I HDACs (HDAC1/2/3), which have also been linked to T-cell exhaustion [111–113] and were connected in our p300-centered subnetwork (**Fig 5B**).

Generally, we observed changes in recruitment for all 3 profiled cofactors between acute and chronically stimulated T cells, with p300 exhibiting the most pronounced changes in recruitment by TF families (**Fig 5F** and **S10 Data**): P300 (14 families), HDACs (10 families), KAT2B (10 families). Consistent with loss of TCR signaling in the exhausted state, we observed loss of cofactor recruitment to IRF/STAT and NF-kB family (REL, NFKB) TFs. Notably, we also observed coherent enhanced recruitment of all 3 cofactors to the TBX family, which contains the major exhaustion regulator EOMES. Strikingly, however, the strongest recruitment signal for p300 in exhausted cells is the NFAT family.

Furthermore, we examined changes in TF-cofactor interactions related to inhibitor receptor expression. We observed increased recruitment of HDACs to PKNOX/TGIF family, both known regulators of PD-1 [114,115]. We also observed p300 recruitment to multiple nuclear receptors families (including RXRA) that are implicated as regulators of TIM-3 [116]. We likewise observed increased binding of KAT2B to another nuclear receptor group, including family member NR4A1 reported to regulate TIM-3, and possibly other inhibitory receptors [117–119]. Increased recruitment of HDACs to the MEIS/PBX family was also seen in the exhausted cells, where MEIS2 is predicted to control TIM-3 [117].

Lastly, we observed enhanced recruitment of p300 to EBOX family in chronically stimulated cells. Of the 3 TFs in this motif family (XBP1, CREB3, CREB3L1), our proteomics data had shown significant changes in XBP1 protein abundance in exhausted cells. Interestingly, XBP1 has been shown to suppress antitumor activity of T cells [120], promote PD-1 expression, and contribute to translating metabolic remodeling to changes in gene expression [12]. Specifically, XBP1 regulates mitochondrial activity, and can be activated by a decrease in glycolysis, as seen in exhaustion.

Together, these results demonstrate changes in TF-cofactor complexes as T cells progress to the exhausted state, supporting our proposed role for p300 in concert with that of known regulators (e.g., EOMES) that likely dictate the metabolic shifts exhibited by exhausted CD4+ T cells.

## Chronic stimulation of CD4+ memory T cells causes cell-intrinsic metabolic remodeling

Our proteomic profiling and network modeling revealed changes in metabolic pathways, as discussed in our pathway enrichment and network analyses results. Due to the large role metabolism plays in T-cell function and differentiation, we performed a systematic metabolic enrichment analysis. Such measurements quantify potential metabolic alterations that associate with chronic stimulation under nutrient rich culture conditions, allowing us to characterize

intrinsic cellular responses rather than secondary changes due to changing nutrient availability. Applying the network inference tool MOMENTA [121] to our proteomic data, we observed significant up- and down-regulation of key metabolic pathways when we compared late exhausted cells to resting cells (**Fig 6A**). We observed changes associated with the electron transport chain and glycolysis in late exhaustion (day 12), providing further evidence of down-regulation of these energy sources in a nutrient-independent manner.

We also observed several pathways pointing to increased, possibly dysfunctional, fatty acid metabolism. Mitochondrial fatty acid oxidation itself is not significantly up-regulated; however, peroxisomal FAO is, along with carnitine shuttling (a key step in importing fatty acids into the mitochondria) and metabolism of omega-3 and omega-6 fatty acids. Peroxisomal FAO serves a different biological function than mitochondrial FAO; peroxisomal FAO is required to break down very long chain fatty acids, prostaglandins, leukotrienes, and other more complex fatty acids. While carnitine does not play a role in peroxisomal FAO, it is necessary to transport end products into the mitochondria for use in the citric acid cycle and OXPHOS. Another difference is where spare electrons are fed during oxidation, which normally goes into the ETC, but peroxisomes donate theirs directly to oxygen ($O_2$) in the cell, which creates reactive oxygen species (ROS). It is known that exhausted T cells often have increased oxidative stress, usually associated with mitochondrial dysfunction; however, this new mechanism may also be contributing. One common way cells adapt to this oxidative stress is through production of antioxidating pathways, such as selenoamino acid metabolism (**Fig 6B**), which is seen up-regulated and known to impact T-cell function via its antioxidant properties [122].

Other top pathways include phosphatidylinositol phosphate and hyaluronan metabolism. Phosphatidylinositol phosphate metabolism, known for generating second messengers with important roles in T-cell function and development [123], plays an important role in the PI3K pathway driving T-cell activation. The down-regulation we observed suggests a possible mechanism underlying exhausted T-cell hyporesponsiveness. Hyaluronan synthesis, also down-regulated, is required for IL-2-mediated cell proliferation [124]. IL-2 secretion by activated CD4 + T cells is rapidly down-regulated by chronic stimulation in our culture model (**Fig 1C**). The down-regulation of hyaluronan would make chronically stimulated cells less responsive to proliferation-inducing cytokines like IL-2.

Consistent with FAO modulation, the most significant hit up-regulated in late exhaustion (day 12) relative to pre-exhaustion (day 6) is lysine metabolism, which along with methionine, is needed for carnitine biosynthesis, as well as 3-oxo-10R-octadecatrienoate beta-oxidation. These results again point to exhausted cells utilizing FAO as a preferential energy source, as previously suggested [11].

We also observed ROS detoxification, which is expected and shares some overlap in functionality with selenoamino acid metabolism. We also observed new pathways emerge, including prostaglandin formation, further encouraging our observation of "C20 Prostanoid Biosynthesis" from our cluster enrichment analysis, and possibly related to peroxisomal FAO, where prostanoids are broken down for energy. Specifically, we see prostaglandin formation from linoleic acid (linoleate), and we also observe up-regulated linoleate metabolism. Inversely, we observed significant pathways that were down in day 12/up in day 6, including glycolysis, pyruvate metabolism, and metabolic pathways of nucleic and amino acids, likely as day 6 cells were still activated and proliferating.

## Direct measurement of metabolite dynamics verifies metabolic remodeling

The putative large-scale alterations in metabolic pathway activity that were inferred by our functional enrichment analysis prompted us to verify these predicted changes using an

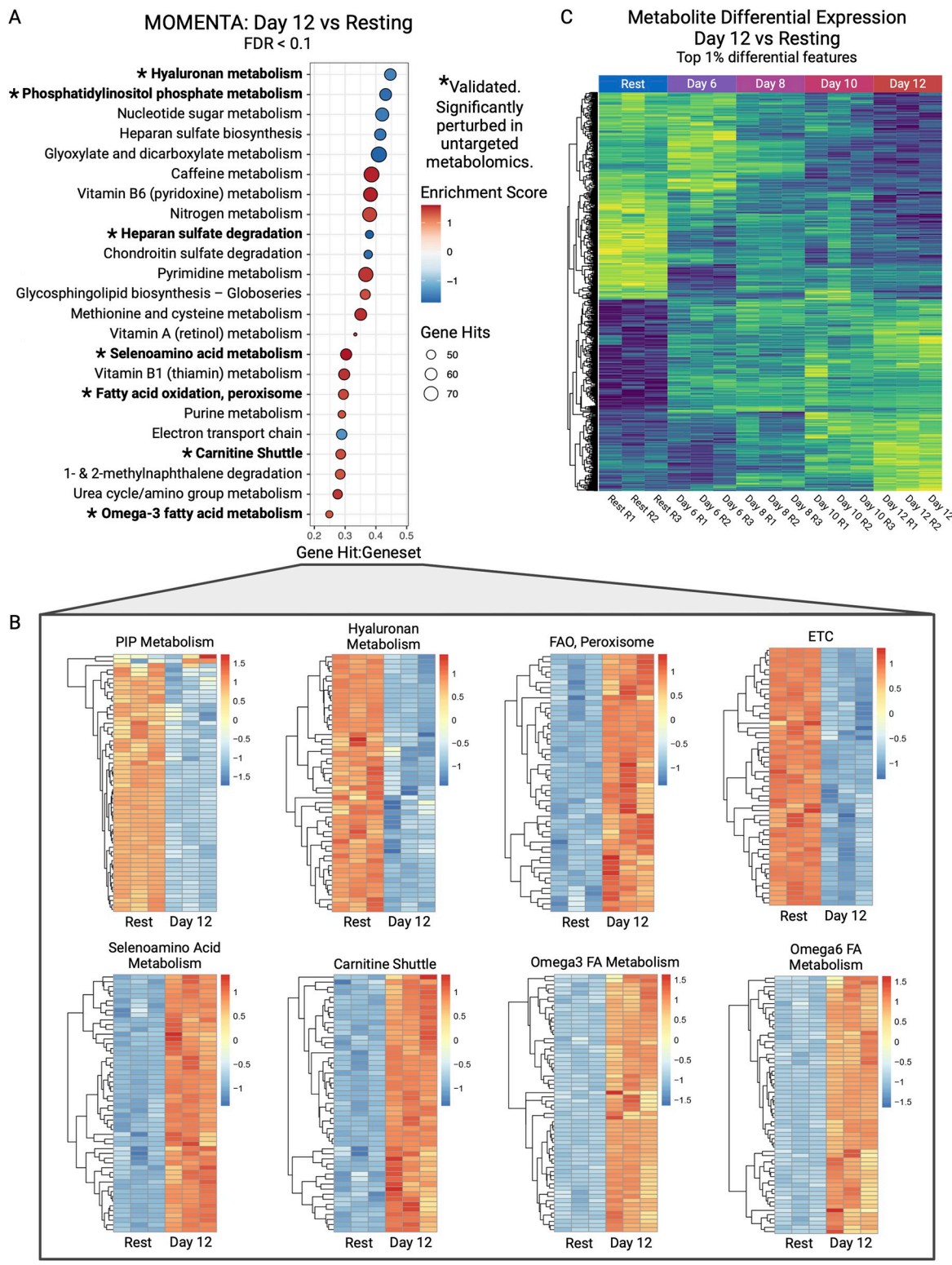

**Fig 6. Metabolic pathway changes in late exhaustion (day 12) chronically stimulated CD4+ T cells.** (A) Metabolic pathway enrichment predictions of proteomic profiles obtained from late exhaustion (day 12) stimulated cells versus resting cells. (B) Heatmaps of expression levels of proteins annotated to individual predicted pathways. Rows z-score normalized. (C) Heatmap of top 1% differential metabolite features measured between late exhaustion (day 12) stimulated and resting cells. Rows z-score normalized. The data underlying this figure can be found in S3 Data (A, B) and S1 Data (C). Figure created with BioRender.com.

independent experimental approach. To this end, we performed global metabolic profiling using untargeted mass spectrometry to directly assess changes in the levels of key metabolites in these same cell populations (see **Methods**). Large changes in the steady-state levels of endogenous cellular small molecules were observed over the exhaustion time course (**Fig 6C**), reflecting profound changes in cellular metabolism.

Consistent with our proteomic predictions, metabolite level functional enrichment confirmed our MOMENTA predictions, as well as implicated other metabolic alterations. Comparing our differential metabolite patterns to the human MFN metabolomic model (curated combination of KEGG, BiGG, and the Edinburgh model) [125], we observed significant perturbations ($p$-value $<0.05$) in many of the pathways that were altered in exhausted cells based on our proteomic results. These include carnitine shuttling, selenoamino acid metabolism, phosphatidylinositol phosphate metabolism, and anti-inflammatory metabolites, such as prostaglandin formation from arachidonate, and EPA. Arachidonate, also known as arachidonic acid, is a precursor to prostaglandin and leukotriene formation.

Hence, our orthogonal approach strongly supports and further refines our findings that chronically stimulated cells intrinsically down-regulate glycolysis and ETC, while breaking down longer/more complex fatty acids through peroxisomal FAO, and up-regulating pathways countering the increased oxidative stress from this altered metabolic state.

## Metabolic pathway regulation in inhibitory-receptor positive HIV-donor T cells reflect metabolic shifts seen in vitro

To validate that the changes observed in the in vitro culture system mirror what occurs in vivo, we obtained T cells from 4 HIV+ donors, which were pooled to obtain enough material for downstream analyses. Cells were FACS-sorted based on inhibitory receptor (IR) expression (PD-1+ and TIGIT+) and CD4/CD8, and bottom-up proteomics was performed. Due to low cell count post-sort, proteomic coverage was shallow, again highlighting the need for the cell culture model. Yet while most exhaustion-associated proteins were not detected, a few (NFATC2, RUNX3, TBX21) were found to be up-regulated in CD8+ IR+ T cells, while others such as DMNT1, were prominently elevated in CD4+ IR+ T cells.

Although certain exhaustion-associated proteins were not directly detected, pathway enrichment analysis using MOMENTA showed high agreement of top metabolic pathways up-regulated in HIV+ IR+ T cells with those altered in day 6 and day 12 (versus resting) in our in vitro model (**Table 1**). These include elevation in pathways related to adenine and adenosine salvage pathways and retinoate biosynthesis.

We also investigated other pathways enrichment encompassing the top 100 most significant factors up-regulated in HIV+ IR+ T cells and found that components related to prostaglandin synthesis, fatty acid metabolism, and p53 pathways were all significantly enriched in exhausted T cells in vivo (**Table 2**), again matching what was seen in our in vitro exhaustion model.

## Up-regulation of exhaustion and identified genes in breast cancer tumor lymphocytes in vivo

To ensure the exhaustion protein expression we observed in our in vitro model also occurs more generally in CD4+ T cells in vivo, we accessed a publicly available single-cell RNA sequencing dataset and plotted marker expression (using Single Cell Portal [126]) in tumor infiltrated CD4+ and CD8+ T lymphocytes displaying an exhausted phenotype (**Fig 7A**). Strikingly, in this breast cancer data set [127], we observed up-regulation of a significant number of exhaustion-associated factors detected in our model (**Fig 7B**), most notably in a putative exhausted CD4+ subpopulation demarcated by GNG4 expression and increased inhibitory

**Table 1. MOMENTA predicts metabolic pathways enriched in HIV+ donor IR+ T cells that reflects many seen in in vitro exhausted cells.**

| Metabolic pathways enriched in IR+ vs. IR- HIV + donor T cells and day 6 vs. resting in vitro exhaustion model | Metabolic pathways enriched in IR+ vs. IR- HIV + donor T cells and day 12 vs. resting in vitro exhaustion model |
| --- | --- |
| ADENINE AND ADENOSINE SALVAGE VI | ADENINE AND ADENOSINE SALVAGE VI |
| GDP-MANNOSE BIOSYNTHESIS | GDP-MANNOSE BIOSYNTHESIS |
| GALACTOSE DEGRADATION I | GALACTOSE DEGRADATION I |
| GLYCOGEN DEGRADATION II | GLYCOGEN DEGRADATION II |
| HYPUSINE BIOSYNTHESIS | HYPUSINE BIOSYNTHESIS |
| ADENINE AND ADENOSINE SALVAGE I | ADENINE AND ADENOSINE SALVAGE I |
| RETINOATE BIOSYNTHESIS I | RETINOATE BIOSYNTHESIS I |
| ARGININE DEGRADATION I | ARGININE DEGRADATION I |
| CITRULLINE DEGRADATION | |
| <IMYO</I-INOSITOL BIOSYNTHESIS | |

MOMENTA was used to perform metabolic pathway enrichment of HIV+ donor IR positive cells against IR negative cells. Shown in the table are significant metabolic pathways from this analysis that are also found in day 6 vs. resting comparison of our data set (left column) and day 12 vs. resting of our data set (right column).

receptor expression (**Fig 7A**, left red box on UMAPs). Inhibitory receptors showed similar levels to that of exhausted CD8+ T cells (**Fig 7A**, right red box on UMAPs). Receptors such as PD-1, TIGIT, and LAG-3, as well as exhaustion-associated transcriptional factors TOX, BLIMP-1 (PRDM1), and NFATC1, among others, were up-regulated in vivo in agreement with our model.

Next, we looked to see if transcripts corresponding to proteins identified in our analysis were likewise expressed in the same CD4+ cell subpopulation. Indeed, we found that despite low general global correlation between mRNA and protein levels that many of the novel markers we identified in our in vitro model were also up-regulated in the tumor resident T cells, including p300, ZEB2, VDR, and many SLC transporters (**Fig 7B**).

Overall, this data suggests that many of the proteins we identified in the chronic T-cell stimulation model are likewise elevated in vivo in exhausted CD4+ (and often simultaneously in exhausted CD8+) T cells in a tumor context.

## Discussion

T-cell exhaustion is a clinically important cell state that many new therapies hinge upon. Full mechanistic understanding of its markers, drivers, and metabolic states are key to developing

**Table 2. Key pathways highlighted in in vitro exhaustion model show enrichment in HIV+ donor IR+ T cells.**

| Pathway | p-Value | Odds ratio | Database |
| --- | --- | --- | --- |
| Prostaglandin biosynthesis and regulation | 0.00001669 | 30.67 | BioPlanet2019 |
| Prostaglandin Synthesis and Regulation WP98 | 0.00007489 | 20.18 | WikiPathway 2023 |
| Hypoxia | 0.003324 | 5.32 | MSigDB Hallmark 2020 |
| p53 Pathway | 0.01807 | 4.19 | MSigDB Hallmark 2020 |
| Fatty Acid Metabolism | 0.04489 | 3.94 | MSigDB Hallmark 2020 |

Selected pathways from Enrichr of the top 100 genes up-regulated in the comparison of IR+ versus IR- HIV donor T cells are shown here as confirmation that these pathways are seen both in our in vitro model as well as in vivo.

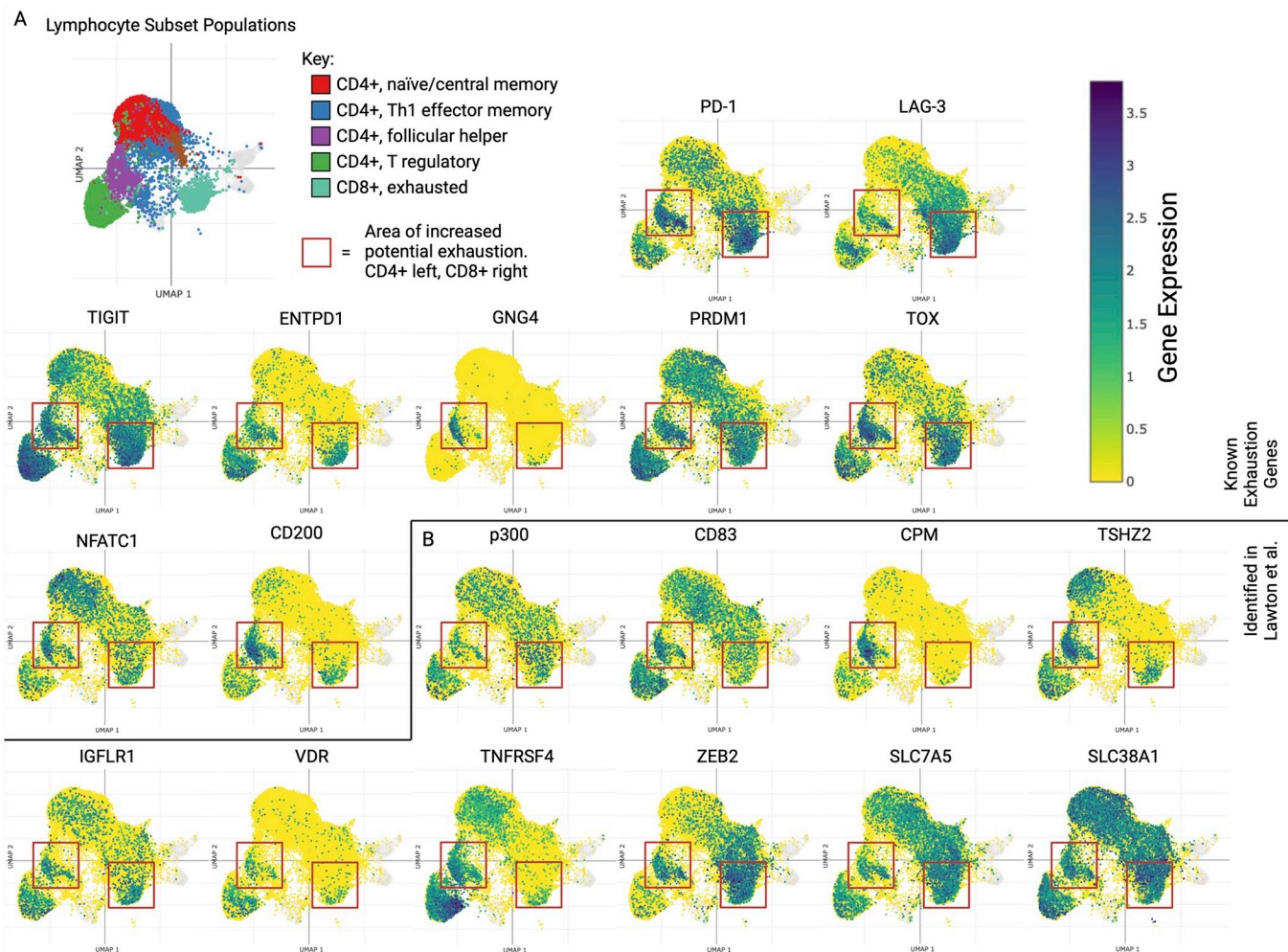

**Fig 7. Gene expression of exhaustion markers and novel proteins identified in exhaustion model using a breast cancer single-cell sequencing data set.** UMAPs of CD4+ subtypes and CD8+ T exhausted tumor lymphocytes showing gene expression of (A) exhaustion-associated genes, and (B) gene expression of proteins identified in the model presented by Lawton and colleagues. UMAPs made using Single Cell Portal, Tarhan and colleagues. The data from this figure and tool used to make it can be found at https://singlecell.broadinstitute.org/single_cell/study/SCP1039/a-single-cell-and-spatially-resolved-atlas-of-human-breast-cancers. Figure created with BioRender.com.

safer, more effective immunotherapies. While considerable work has been done in characterizing the transcriptional and epigenetic components governing T-cell exhaustion, most of the work to-date has focused on exhaustion in the context of CD8+ effector T cells, lacking a full characterization of this phenotype in a multiomic fashion shown here. Moreover, most studies to date have utilized RNA-sequencing to characterize exhausted T cells, we sought to characterize their protein expression since mRNA transcript levels do not correlate well with protein expression, especially in CD4+ T cells [128].

To address these gaps, we profiled chronically stimulated CD4+ T cells using proteomic methods and identified candidate markers/drivers. Hierarchical clustering revealed expression profile patterns enriched for inhibitory receptors (PD-1, TIM-3, TIGIT, LAG-3, etc.) and transcriptional factors (TOX, EOMES, ID2, etc.), while implicating other novel exhaustion markers and potential causal regulators.

From these clusters, several proteins are of notable interest. Specifically, receptors CD276 and FLT1 have not previously been linked to CD4+ T-cell exhaustion. However, their

annotated functions as inhibitory receptors or family of regulators of TOX, respectively, suggest they play roles in dampening T-cell responses and driving exhaustion. Likewise, while not been previously linked to CD4+ T-cell exhaustion, the zinc finger homeobox-binding transcription factor ZEB2 was previously observed up-regulated in other chronically stimulated CD4+ T-cell populations [99], although its exact role requires follow up.

Our integrative modeling pointed to the epigenetic regulator p300 as a promising lead, which we evaluated using small molecule inhibitors. In other contexts (e.g., cancer, cell cycle control), p300 is a well-studied epigenetic regulator, but has not been implicated as a regulator in CD4+ T-cell exhaustion yet. It is known that p300 binds to and regulates NFAT family proteins, and so some of its effects are likely reflected through this relationship. However, we also observed increased binding of p300 to other transcription factor motifs, which may modulate the exhaustion phenotype outside of its control of NFATs. Despite the decrease in PD-1 and TIM-3 expression found upon inhibition of p300, we did find an increase in other inhibitory receptors, namely LAG-3 and TIGIT. This suggests that targeting p300 directly may not have much direct effect on T-cell exhaustion or function, as the increased LAG-3 and TIGIT could cancel out effects of decreased PD-1 and TIM-3, or even potentially exacerbate the phenotype. Follow up studies to find intermediate factors that may play a role in linking p300 to PD-1 and LAG-3 expression would be beneficial to circumvent the increase in other inhibitory receptors.

Lastly, our resource provides insights into the large metabolic shifts previously reported only for exhaustion of CD8+ T cells, including up-regulation of SLC nutrient transporters. Our observed metabolic remodeling is likely driven in part by p300 and some of its interacting transcription factors; for example, EBOX proteins like XBP1, which regulates lipid metabolism and its transcriptional activity, has been linked to metabolic dysfunction [129]. We also predicted and independently verified down-regulation of glycolysis and electron transport chain for energy production in exhausted cells. While such a metabolic shift in dysfunctional T cell in the tumor microenvironment has been reported before [12], our results suggest that some of these changes are immune cell-type intrinsic rather than a consequence of nutrient-deficient microenvironment since cells in our model were provided with continuous access to nutrients (fresh medium provided daily).

One advantage of using our culture model of chronic stimulation is that cells are not interacting with a tumor microenvironment, which is usually starved of nutrients, allowing us to uncouple cellular responses to chronic stimulation from effects of the tumor microenvironment. It remains unclear whether metabolic changes observed in chronically stimulated/ exhausted T cells are due to this nutrient-lacking environment or is a cell-intrinsic response. However, our analysis indicates that metabolic dysfunction occurs even when in the presence of the required nutrients. Strikingly, we observed profound changes in fatty acid oxidation, whereby exhausted cells showed marked up-regulating peroxisomal (but not mitochondrial) FAO pathways. We found up-regulation of Omega-3 and -6 fatty acid metabolism, along with enhanced prostanoid metabolism, indicating a major shift in how cells process lipids. Recently, (CD8+) T cells with PD-1$^{high}$ within non-small-cell lung carcinoma were observed to have greater lipid content than the PD-1$^{low}$ subpopulation [12,130]. Combined with our findings of increased peroxisomal FAO (but no change in mitochondrial FAO), such an observation implies lipid accumulation upon partial breakdown. Notably, exhausted T cells have increased oxidative stress caused by ROS [131], which was previously attributed primarily to impaired mitochondrial function. Our observation of increased peroxisomal FAO, which donates spare electrons to free cellular $O_2$ instead of feeding them into the electron transport chain as in mitochondrial FAO, may plausibly contribute to increased ROS. However, to what degree it contributes is currently unclear and requires future study.

Intriguingly, we observed up-regulation of selenoamino acid metabolism, possibly as a response to increased ROS, which has not previously been reported. Selenium is a trace essential element incorporated into amino acids in place of Sulphur, such as in cysteine and methionine, to provide direct anti-oxidant effects even prior to incorporation into antioxidant proteins [132].

Although exhibiting extensive similarities in inhibitory receptor expression as observed in CD8+ T-cell exhaustion, we did not see complete agreement in that common inhibitory receptors CTLA-4 and CD44 were not up-regulated in our CD4+ data set. CTLA-4, an inhibitory receptor previously reported as a marker of exhausted T cells is the target of blockade therapies [133–135]. Perhaps the lack of this factor is characteristic of the chronically stimulated CD4 + T-cell phenotype we achieved with our in vitro model, or the protein surface expression may be regulated mainly by localization, which we did not test for in our flow cytometry panel. CD44 is known to be expressed on memory cells after exposure to antigen [136], which we confirmed as up-regulated in resting memory cells, but fails to return to baseline in our chronic stimulation time course.

Also notable was up-regulation of IFN-γ at the protein expression level, in contrast to its impaired secretion in response to antigen exposure by day 12, whereupon it drops to undetectable levels. Increased IFN-γ is not uncommon in cultured CD4+ T cells, but the mechanism is unclear. It is possible that the degree of chronic exposure to antigen in our model selects for cytotoxic CD4+ lymphocytes, a durable subtype originally thought to be an artifact of long-term cell culture but brought to the spotlight in recent years for its important role in fighting chronic infections [137].

## Study limitations

While our culture model allowed for in-depth multiomic profiling and nutrient-independent observations, it does have some limitations. The in vitro conditions (nutrient availability, cell–cell interactions, etc.) and method of continuous TCR activation (via CD3/CD28 Dynabeads) are artificial and do not fully replicate what immune cells experience in vivo. However, the up-regulation of many known exhaustion markers and regulators strongly supports the utility of the model. In order to more closely replicate physiological conditions, human serum was used in place of fetal bovine serum. We also recognize these results were largely reproduced in a small number of human donors, and so may not generalize more broadly due to genetic and environmental variability.

Another confounder is cellular heterogeneity, given that bulk omics methods were used here, as T-cell populations are often heterogeneous. In particular, we observed evidence of variability by flow cytometry based on co-expression levels of inhibitory receptors. Although our quantitative mass spectrometry-based approach measures the average expression across all the subpopulations, we believe our results captured trends that are representative across the entire population. With advances in proteomics, phosphoproteomics, and metabolomics methods, the ability to further hone in on specific subpopulations will one day be feasible and make for interesting follow-up studies.

We also acknowledge that our phosphoproteomics method detected primarily phosphorylation sites that have not yet been functionally annotated. While lack of curated functional significance and incomplete kinase-substrate knowledge are current bottlenecks in this field. However, the data it is still insightful as signaling changes were predicted at the pathway level, and observed enriched phosphorylation of proteins in the same system serves as an indicator of differentially regulated activity.

In closing, this work provides a massive resource of multiple data types temporally profiling the effects of CD4+ T-cell exhaustion/chronic stimulation for the broader research community. Our data allows comparison to existing results and ongoing in vivo studies, points to potential new targets and avenues of research, including the identification of regulatory factors that can potentially be modulated to either reduce exhaustion or potentially even to induce the exhausted state in hyperactive contexts such as autoimmune disease. With the major findings of our study and the limitations in mind, this resource opens the door for follow-up studies. Further research into the mechanisms of control of p300 during chronic T-cell activation should permit dissection of the relationships to other important upstream and downstream proteins. Such analyses could identify new druggable targets with a more specific mode of action as targeting p300 itself is detrimental to normal cell function due to cell-cycle interference. Likewise, as metabolism dictates T-cell function, deploying tool compounds to investigate the impact of the metabolic changes we detected, most notably preferential alterations in peroxisomal fatty acid oxidation, and the associated regulators driving these alterations, could reveal actionable targets.

## Methods

### Human primary PBMC isolation

Blood was obtained through New York Biologics, Inc. It is collected from de-identified healthy donors and shipped refrigerated directly to our lab. The blood is tested for potentially dangerous viral or bacterial infections. PBMCs were isolated from blood packs using density centrifugation with Histopaque-1077 (Sigma-Aldrich, 10771). PBMCs are washed multiple times with PBS before being frozen in 10% DMSO + 90% HI-FBS (HyClone) via Mr. Frosty container placed at $-80°C$ overnight, and then transferred to long-term liquid nitrogen storage. All data produced from these samples were analyzed anonymously.

### T-cell isolation

Frozen PBMCs were thawed quickly at 37°C in a water bath until just an ice core remained. Warm media was added and then returned to a 15 ml Falcon tube until PBMCs were thawed, and then spun down. After aspiration, cells were resuspended in PBS with 1% BSA (Thermo, 15260037) and 50 mM EDTA, and T cells were isolated using untouched memory CD4+ T-cell isolation kit (Miltenyi, 130-091-893). T cells were then plated on appropriately sized tissue culture plates in AIM-V (Thermo, 12055091) with 5% heat-inactivated human AB serum (Valley Biomedical, HP1022HI) and incubated at 37°C (5% $CO_2$) for 2 h to remove any contaminating monocytes. Cells in suspension were then collected and used for expansion or experiment.

### T-cell expansion

To obtain enough cells/material for mass spectrometry-based approaches, T cells were expanded before use. After T-cell isolation, T cells were plated in AIM-V with 5% heat-inactivated human AB serum (hereby referred to as T-cell media) at a density of $1 \times 10^6$ cells per ml. For CD4+ T cells, 30 U/ml of IL-2 was added, and 25 μl of ImmunoCult (CD3/CD28, Stemcell Technologies) was added per million cells. On day 3, cells were diluted approximately 4- to 8-fold for a concentration of $1–2.5 \times 10^5$ cells per ml. On day 5, cells were again diluted to a concentration of $1–2.5 \times 10^5$ cells per ml. On day 7, cells were diluted to as close to 4-fold as possible to achieve between $1–6 \times 10^5$ cells per ml. Cell counts and viability were then

monitored each day until day 9 or 10 and were harvested when proliferation rate and viability began to decline. These cells were then used for experimental procedures.

## T-cell in vitro exhaustion

Once isolated and expanded, T cells were plated at $1 \times 10^6$ cells per ml, with IL-2 added where specified. CD3/CD28 Dynabeads (Thermo, 11132) were added at a 1:1 cell to bead ratio. On day 2, cells were collected, spun down, resuspended in 1 ml of T-cell media and beads were removed using a magnet. Beads were then washed 2 more times, each with more agitation than the last to dissociate any cells stuck to beads. For "resting" condition samples, cells were then plated in fresh medium at $1 \times 10^6$ cells per ml, and for "chronic" stimulation samples, cells were re-plated at the same density, but had fresh Dynabeads added at 1:1. Media and fresh beads were then added every 2 days to the chronic stimulation samples until end of time course (day 12).

## Small molecule inhibition cultures

Cells were cultured through the in vitro exhaustion protocol described above until day 10. After the day 10 media and antigen refresh, cells were treated with 1 to 10 μm of A-485 p300 inhibitor, or with Dasatinib to block T-cell activation. "Rested" condition cells were also plated without Dynabeads to allow for 2 days of rest before analysis. All conditions were then cultured for 2 more days before analysis via flow cytometry.

## Cytokine release assay

On time points where cytokine release was tested, $5 \times 10^5$ cells were plated in triplicate per replicate in 1 ml of media without IL-2 in a 12-well plate. CD3/CD28 Dynabeads were added at 1:1. Cells were incubated for 6 to 10 h before media was collected and frozen at −80°C. Once all time points were acquired, media samples were thawed and analyzed via ELISA (Thermo, BMS221INST, KHC3011, EHIFNG) or Proteome Profiler Human Cytokine Array Kit (R&D Systems, ARY005B).

## Mass spectrometry sample preparation

Samples were taken from −80°C storage and immediately lysed using guanidine hydrochloride lysis buffer (6 M guanidine hydrochloride, 100 mM (pH 8.5) Tris, 40 mM chloroacetamide, 10 mM TCEP, PhosStop). Bradford was performed on each sample to obtain protein concentrations. Protein mixtures were then enzymatically digested using trypsin protease at a 1:50 to 1:100 μg of trypsin to protein ratio. Digestion was performed overnight (16 h) at 37°C with gentle shaking. Samples were then acidified using final concentration of 1% formic acid to stop digestion. Each sample was then desalted using Sep-Pak desalting columns and dried down using a speed-vac before peptide concentration was measured using Pierce peptide quantification assay (Thermo, 23275). Approximately 100 μg of each sample was then labeled using TMTpro 16-plex kit (Thermo, A44520). Samples were then pooled after labeling and dried down in a speed-vac. Next, the pooled samples were resuspended in 2% acetonitrile, 0.1% ammonium hydroxide and loaded onto the HPLC (Agilent 1100). Reversed-phase chromatography was performed with a 53-min gradient, where 48 fractions were collected starting after 5 minutes of loading. Every 12th fraction was then pooled, and dried down, for a final total of 12 fractionated samples; 5% of these were taken and used for proteomic analysis, and the rest was used for phosphopeptide enrichment and subsequent phosphoproteomic analysis.

## Phosphopeptide enrichment

Fractionated and dried samples were thawed after storage at −80˚C on the bench for 5 min. Each fraction was then resuspended in 800 μl of sample buffer (80% acetonitrile, 0.5% trifluoroacetic acid) and spun at 1,600 rpm on a temperature-controlled shaker at 20˚C for 15 min. Resuspended fractions were then loaded onto 96-well plates and placed in the KingFisher (Thermo Fisher) sample purification system. Six other plates were prepared and placed into the KingFisher alongside the sample plate: (1) Plate with comb tip to cover magnetic rods. (2) Bead plate, containing washed iron beads (10 μl beads/100 μg of peptide, Cube Biotech, 31505-Fe) in wash buffer (80% acetonitrile, 0.1% trifluoroacetic acid). (3) First wash plate containing 800 μl of wash buffer. (4) Second wash plate containing 800 μl of wash buffer. (5) First elution plate containing 200 μl of elution buffer (50% acetonitrile, 7.5% ammonium hydroxide). (6) Second elution plate containing 200 μl of elution buffer. The KingFisher was programmed to pick up the iron beads using the magnetic rods and incubate them with sample for 20 min with agitation. Beads (now with bound phosphopeptides) were then transferred to the first wash plate and incubated for 5 min with agitation, and then again for the second wash plate. Finally, beads were transferred to the elution plates and incubated for 5 min with agitation each. The 2 eluates for each fraction were then collected and pooled before drying down in a SpeedVac.

## Metabolite extraction and clean-up from cellular supernatant

If samples were intended for metabolomics in parallel with proteomics, samples were not immediately lysed with guanidine lysis buffer as described above. Instead, samples were snap frozen after washing with PBS in a chemically resistant microcentrifuge tubes (e.g., Eppendorf). A mixture of pre-chilled (−20˚C) methanol/acetonitrile/water (40/40/20, v/v) was prepared for liquid–liquid extraction (LLE). Next, we added buffer at 3 to 5× volume of sample. Samples were then vortexed for 30 s and snap frozen by placing in liquid nitrogen until frozen (1 min). Samples are thawed on ice and sonicated for 3 cycles of 5-min pulses using a probe sonicator (40 kHz, Ultra Autosonic) at 10% power. The samples were then incubated at −20˚C for 1 h. Samples were next centrifuged for 15 min at max speed (14,000 rpm) at 4˚C. The supernatant containing metabolites were transferred to a new tube, while protein (precipitate) was kept at −80˚C until ready, at which point guanidine lysis buffer was added and brought through the proteomic workflow. Metabolite supernatant was dried in a speed-vac at 30˚C prior to solid phase micro-extraction (SPME) clean up.

Dried metabolite extracts were resuspended in PBS and transferred to a 96-well plate. The 96-blade SPME system was next washed in ethanol:water (70:30, v/v) for 30 min and preconditioned for 30 min in methanol:water (50:50, v/v). Samples were then extracted for 1 h, and the blades with bound metabolites were washed for 20 s in water (MS grade, Thermo Fisher) and metabolites desorbed in acetonitrile:water (50:50,v/v) for 1 h. The desorption solvent was then evaporated in a speed-vac until dry and stored at −80˚C until ready for analysis prior to LC/MS, at which point it was resuspended in 2% acetonitrile and separated on the nano-liquid chromatography mass spectrometer (nLC-MS/MS).

## LC-MS analysis of peptides and phosphopeptides

Fractionated, multiplexed samples were resuspended in 2% acetonitrile and 0.1% formic acid (mobile phase A) and transferred into a 96-well plate to be loaded into our Easy nanoLC1200 HPLC system and analyzed by our Exploris 480 with FAIMS Pro (Thermo Fisher). The peptide solutions were first loaded onto a reversed-phase nanotrap column (75 mm i.d. 3 2 cm, Acclaim PepMap100 C18 3 mm, 100 A˚, Thermo Fisher) and separated by an EASY-Spray column (ES803A, Thermo Fisher) using a gradient (6% to 17% over 77 min, then 17% to 36%

over 45 min for peptides, 6% to 19% over 58 min, then 19% to 36% over 34 min for phospho-peptides) of mobile phase B (0.1% formic acid, 80% acetonitrile) at a flow rate of 250 nL/min-ute. Positive ion mode was used throughout, with a capillary temperature of 275°C and a spray voltage of 2,100 V. Our DDA method chose the 12 most abundant ions for fragmentation (Exploris 480 NCE 33%), with FAIMS cycling through −50, −57, and −64 volts for each full scan. Precursor scans were acquired at a 120,000 FWHM resolution with a maximum injection time of 120 ms in the Orbitrap analyzer. The following 0.8 s were dedicated to fragmenting the most abundant ions at the same FAIMS compensation voltage, with charge states between 2 and 5, via HCD (NCE 33%) before analysis at a resolution of 45,000 FWHM with a maximum injection time of 60 ms. Phosphopeptides were analyzed the same besides the shortened gradi-ent described and increased injection time of 150 ms.

## LC-MS analysis of metabolites

LC-MS was performed using an Orbitrap Q-Exactive Exploris 480 mass spectrometer (Thermo Scientific). Metabolites (2 μls) were loaded onto a pre-column (75 mm i.d. × 2 cm, 3 μm) and then separated on a C18 capillary column (75 mm i.d. × 2 cm, 2 μm, 100 Å, Thermo Fisher Scientific) over a 45-min gradient. Mobile phase A was 2% acetonitrile and mobile phase B was 80% of acetonitrile. The gradient consisted of 2% to 60% mobile phase B for 20 min, was increased to 95% mobile phase B over 10 min, and maintained at 95% mobile phase B for 15 min with flow rate of 300 nL/min. The MS was operated in switching mode to acquire data on both positive and negative mode in the same run. The spray voltage was set to 2.1 kV and −1.8 kV, respectively. Full scan covers m/z 60 to 1,000 at a resolution of 60,000. The AGC target was set to 300% and the maximum ion injection time was set to 25 ms. MS2 scans were performed at 15,000 resolution with a maximum injection time of 64 ms using stepped colli-sion energies of 10, 20, and 40. Dynamic exclusion was enabled using a time window of 10 s.

## Prize-collecting Steiner forest

The Steiner forest problem is a method of network optimization in a weighted graph $G(V, E)$ that is composed of a node set $V$ and edge set $E$. Furthermore, we consider a function $p(v)$ that assigns a prize to each node $v \in V$ and a function $w(e)$ that assigns a weight to each edge $e \in E$. Formally, we seek a forest of trees $F(V_F, E_F)$ that minimizes the objective function

$$f(F) = \sum_{v \notin V_F} \beta \cdot p(v) - \mu \cdot degree(v) + \sum_{e \in E_F} c(e) + \omega \cdot \kappa c(e) + \omega \cdot \kappa,$$

where $c(e) = 1 - w(e)$, $\beta$ is a scaling factor that affects the number of prized nodes included in the optimal forest, $\mu$ is a parameter that penalizes hub nodes (nodes with high degree), $\kappa$ is the number of trees in the forest, and $\omega$ is a parameter that controls the number of trees. Prior to computing the optimal forest, a dummy node is attached to a subset of the nodes in $G$. Once the optimization is complete, the dummy node and all of its artificial edges are removed to reveal a forest with each tree in the forest rooted at a node that was connected to the dummy node. We used the Omics Integrator package to solve the Steiner forest problem [138]. As the Steiner forest algorithm requires an input network, we assembled a directed molecular net-work capturing transcription factor–target, kinase–substrate, phosphatase–substrate as well as other interactions from Reactome [139], Kegg [140], Signor [141], and ligand–receptor inter-actions [142].

For each interaction $e$ in the input network, the Steiner forest problem requires the assign-ment of a weight or probability $w(e)$, suggesting that edges with low cost $c(e) = 1 - w(e)$ are more likely to be selected in the optimal forest. Given a directed edge e = (x, y), where the

node $x$ is the tail or the source of the interaction and node $y$ is the head, we defined that the weighting function is the reciprocal of the outdegree of x as $w(e) = k_{out}^{-1}$. Consequently, edges that involve tail genes with high outdegree will be more likely to be removed during the Steiner forest optimization. Such a step allowed us to more effectively penalize hub nodes, indicating higher confidence that edges selected in the optimal forest are more specific to T-cell exhaustion instead.

As for a set of priced genes, we considered genes that were significantly differentially expressed or phosphorylated in each time step after stimulation if their corresponding FDR < 0.05. Furthermore, we assigned a score to each prized gene defined as the absolute values of the $lg_2$ of the corresponding fold changes in a given time point $t$ compared to the unstimulated case, $p_t(v) = \left|\log_2\left(\frac{m_t(v)}{m_0(v)}\right)\right|$.

In addition, Omics Integrator requires the user to assign values to certain parameters that affect the topology of the optimal forest. The parameters of interest are $\mu$, $\beta$, $\omega$, and $D$. We assigned a value of 0 to parameter $\mu$ because we found $\mu$ to be too punitive, and the resulting optimal forest would either be empty or leave out all but a couple of nodes with non-zero prizes, even for small values $\mu \sim 10^{-4}$. In addition, our edge weight-assignment in (1) already punishes hub nodes that might be overrepresented in the literature. Furthermore, we set $\beta = 1$ and $\omega = 1$. Lastly, we assigned a value of 5 to $D$. The values chosen for $\mu$, $\beta$, $\omega$, and $D$ were constant across time points because any differences in network size and structure among the optimal forests could then be attributed to the experimental data and the input networks and not to different parameter values.

## Nuclear extraction

The nuclear extract protocols are as previously described [108,143,144], with modifications as detailed below. Nuclear extractions were performed on day 2 of the exhaustion protocol for acutely stimulated cells and day 12 of the exhaustion protocol for exhausted cells. To harvest nuclear extracts from the stimulated CD4+ memory T cells, cells were collected with media in 50 ml conical tubes and centrifuged at 500×g for 5 min at 4˚C. After centrifugation, the supernatant was aspirated off leaving behind the cell pellet. Cell pellets were resuspended in ice-cold 1× PBS (Cytvia, Catalogue #SH3028.02) and 0.1 mM Protease Inhibitor (Sigma-Aldrich, Catalogue #P8340). If there were multiple cell pellets from the same condition, they were combined into 1 tube at this step. Cells were centrifuged at 500×g at 4˚C and the supernatant was aspirated off. To lyse the plasma membrane, the cells were resuspended in Buffer A and incubated for 10 min on ice (10 mM HEPES, pH 7.9, 1.5 mM MgCl, 10 mM KCl, 0.1 mM Protease Inhibitor, Phosphatase Inhibitor (Santa-Cruz Biotechnology, Catalogue #sc-45044), 0.5 mM DTT (Sigma-Aldrich, Catalogue #4315)). After the 10 min incubation, Igepal detergent (final concentration of 0.1%) was added to the cell and Buffer A mixture and vortexed for 10 s. To separate the cytosolic fraction from the nuclei, the sample was centrifuged at 500×g for 5 min at 4˚C to pellet the nuclei. The cytosolic fraction was collected into a separate microcentrifuge tube. The pelleted nuclei were washed gently with Buffer A, which was added and removed without disrupting the pellet. The pelleted nuclei were then resuspended in Buffer C (20 mM HEPES, pH 7.9, 25% glycerol, 1.5 mM MgCl, 0.2 mM EDTA, 0.1 mM Protease Inhibitor, Phosphatase Inhibitor, 0.5 mM DTT, and 420 mM NaCl) and vortexed for 30 s. To extract the nuclear proteins (i.e., the nuclear extract), the nuclei were incubated in Buffer C for 1 h while mixing at 4˚C on Life Technologies HulaMixer (Orbital: OFF, Reciprocal: 90˚/30 s, Vibro: 5˚/5 s). To separate the nuclear extract from the nuclear debris, the mixture was centrifuged at 21,000×g for 20 min at 4˚C. The nuclear extract was collected in a separate microcentrifuge tube and flash frozen using liquid nitrogen. Nuclear extracts were stored at −80˚C.

## Protein-binding microarray (PBM)

Microarray DNA double stranding and PBM protocols are as previously described [108,143–145]. Any changes to the previously published protocols are detailed. Double-stranded microarrays were pre-wetted in HBS (20 mM HEPES, 150 mM NaCl) containing 0.01% Triton X-100 for 5 min and then de-wetted in an HBS bath. Microarrays were incubated with 2% non-fat dried milk solution (Lab Scientific, Cat # M0841) in HBS for 1 h in the dark. The array was then rinsed in an HBS bath containing 0.1% Tween-20 and subsequently de-wetted in an HBS bath. Next, the array was incubated with nuclear extract (420 μg of nuclear extract for $4 \times 180$ K array design) for 1 h in the dark in a binding reaction buffer (20 mM HEPES, pH 7.9, 100 mM NaCl, 1 mM DTT, 0.2 mg/ml BSA, 0.02% Triton X-100, 0.4 mg/ml salmon testes DNA (Sigma-Aldrich, Catalogue #D7656)). The array was then again rinsed in an HBS bath containing 0.1% Tween-20 and de-wetted in an HBS bath. After the protein incubation, the array was incubated for 20 min in the dark with 20 μg/ml primary antibody for the COF of interest (**S1 Table**). The primary antibody was diluted in 2% non-fat dried milk in HBS. After the primary antibody incubation, the array was rinsed in an HBS bath containing 0.1% Tween-20 and de-wetted in an HBS bath. Microarrays were then incubated with 20 μg/ml of either alexa488 or alexa647 conjugated secondary antibody (**S1 Table**) for 20 min in the dark. The secondary antibody was diluted in 2% milk in HBS. The array was rinsed in bath containing HBS and 0.1% Tween-20. The array was then place in a Coplin Jars twice for 3 min in 0.05% Tween-20 in HBS and once for 2 min in HBS in Coplin jars to remove excess antibody. After the washes, the array was de-wetted in an HBS bath. Microarrays were scanned with a GenePix 4400A scanner and fluorescence was quantified using GenePix Pro 7.2. Exported fluorescence data were normalized with MicroArray LINEar Regression.

Replicate PBM experiments for each COF in both acute and exhausted T-cell conditions were as follows: KAT2B (4 replicates each condition); HDAC mix (HDAC 1, II, III) (4 replicates each condition); P300 (2 replicates each condition). Reported TF-COF interactions in each conditions had to be seen in 2 replicate experiments.

## PBM microarray design and analysis

The microarray design contained a set of 346 nonredundant TF-binding models from the JASPAR 2018 core vertebrate motif set and was obtained using the JASPAR 2018 R bioconductor package. TF-binding models refer to the consensus sequence for each motif and all single-nucleotide variants (SVs) of the consensus, which allows us to directly construct motifs from our binding data. Each transcription factor binding site (TFBS) (i.e., consensus or SV sequences) was embedded within a 34-base pair (bp) long target region, within the context of a 60-bp long DNA probe of the form: GC cap (2 bp) + target region (34 bp) + primer sequence (24 bp). Within the target region, we embedded each consensus sequence with a variable number of padding bases to account for different length motifs, using the format: 5′ pad (2 bp) + consensus sequence (L bp) + 3′ variable pad (32–L bp). The pad sequences (both 5′ and 3′) were generated randomly with the constraint that sequential positions contain non-repeating nucleotides. Background target DNA probes were also included in the design to estimate background fluorescence intensities in the experiments. These regions were selected at 34-base genomic segments from the human genome (hg38).

Cofactor recruitment motifs modeled using single variant (SV) probes was performed as previously described [108,143]. DNA probe log fluorescence values were first z-transformed using the fluorescence values from the background probes. DNA motifs were then determined for all TF-binding models (i.e., sets of consensus and associated SV probe sequences). For each TF-binding model, the base preference at each position was evaluated using Δz-scores

calculated as the difference relative to the median z-score obtained across all possible nucleotides at that position (i.e., the consensus and the 3 SV probes). This z-score-based binding motif captures the differential impact of base variants on the log fluorescence binding, which is analogous to a change in binding energy. The z-score matrices can be transformed into position weight matrices (PWMs) to facilitate comparison against motif databases using the approach previously described [108,143]. Motifs were only considered significant if they passed the following criteria. (1) z-score of the consensus probe for each TF-binding models >0.5; (2) the average information content (IC) of at least one 5-bp window within the motif must > 1; (3) the adjusted $p$-value of the best motif match against JASPAR motif database >0.05. Motif matching was performed using TomTom (Meme suite) [146]. Motif strength is the mean of $\Delta$z-score for the top 15% of probes for each TF-binding model. Individual TF-binding motifs were clustered into TF family groups based on highly similar and generally indistinguishable motifs. Matched motifs were called into their TF family cluster. The motif strength for each TF family group is the mean of all the significant motifs within that TF family cluster.

Antibodies used for these assays can be found in **S1 Table**.

## Fluorescence-activated cell sorting (FACS)

Briefly, ~$2 \times 10^8$ frozen PBMCs from 4 people living with HIV (PWH) and an uninfected person were thawed, washed, and stained with Zombie NIR viability dye (1:400, Biolegend) in phosphate-buffered saline (DPBS) for 15 min at room temperature (RT). PBMCs from the 4 PWH were combined into 1 tube for staining and sorting while PBMCs from the uninfected individual were used for necessary fluorescence experimental controls. Viability staining was quenched with FACS buffer (DPBS + 0.5% bovine serum albumin + 2 mM ethylenediaminetetraacetic acid), PBMCs were washed, and then pre-stained with a Fc receptor blocking solution (human TruStain FcX, 1:50, Biolegend) for 10 min at RT. Next, PBMCs were stained using an 8-parameter antibody cocktail (CD8 (BUV805, SK1, BD Biosciences), PD-1 (BV421, EH12.2H7, Biolegend), CD3 (BV510, OKT3, Biolegend), TIGIT (PE-Cy7, A15153G, Biolegend), CD4 (Alexa Fluor 700, RPA-T4, Biolegend), CD14 (APC/Fire750, 63D3, Biolegend), and CD19 (APC/Fire750, SJ25C1, Biolegend)) combined with monocyte blocker (1:20, Biolegend) and brilliant stain buffer plus (1:10, BD Biosciences) for 30 min on ice. After cell surface staining, PBMCs were washed twice with FACS buffer, resuspended in FACS buffer, and kept on ice until acquisition. All staining was performed in the dark and cells were sorted using a FACSAriaTM cell sorter (BD Biosciences). Full stained PBMCs from PWH were sorted into 4 populations of cells based on IR expression on T cells: CD4+ PD-1+ +/- TIGIT+ (denoted IR+) cells, CD4+ PD-1- TIGIT- (denoted IR-), CD8+/- CD4- (non-CD4 CD3+) PD-1+ +/- TIGIT+ (denoted IR+), and CD8+/- CD4- (non-CD4 CD3+) PD-1- TIGIT- (denoted IR-); ~1–2.5 × 106 cells per population were collected for downstream proteomic analysis. Additional information for all antibodies used for FACS can be found in **S2 Table**.

## Additional demographic information regarding subjects used for sorting

The 4 people living with HIV infection ranged from 26 to 56 years old, included 2 males and 2 females, 2 identified as black/African American, and 2 identified as Hispanic; all 4 individuals were part of the Boston University HIV/Aging cohort with an enrollment requirement of effective viral suppression with undetectable HIV-1 RNA for a minimum of 6 month (<50 copies/ml). The uninfected individual used for fluorescence controls was a 71-year-old white male; cells from this donor were derived from a leukopak blood pack (New York Biologics).

## Data reporting

All relevant data and supplemental data are within the paper and its Supporting Information files excluding raw flow cytometry and mass spectrometry files, which can be found in the following locations: Flow cytometry files are freely accessible at http://flowrepository.org using the identifier "FR-FCM-Z8G6," and the mass spectrometry data files have been deposited to the ProteomeXchange Consortium via the PRIDE partner repository with the data set identifier PXD057703. For more information, please see reference below.

Perez-Riverol Y, Bai J, Bandla C, Hewapathirana S, García-Seisdedos D, Kamatchinathan S, Kundu D, Prakash A, Frericks-Zipper A, Eisenacher M, Walzer M, Wang S, Brazma A, Vizcaíno JA (2022). The PRIDE database resources in 2022: A Hub for mass spectrometry-based proteomics evidences. Nucleic Acids Res 50(D1):D543-D552 (PubMed ID: 34723319).

## Supporting information

**S1 Fig. Second donor time course flow cytometry and proteomic overview.** (A) Flow cytometric analysis of PD-1, LAG-3, TIGIT, TIM-3, CD160, and CD137 of time points (day 0, 2, 6, and 12) of a second donor CD4+ memory T cells brought through the in vitro exhaustion protocol. (B) Proteomic expression of exhaustion factors and selected novel factors to match those shown in main text. The data underlying this figure can be found in the supplementary flow files uploaded to flowrepository.org (A), and in S2 Data (B). Figure created with BioRender.com.
(TIFF)

**S2 Fig. Cell viability overview.** Flow cytometric analysis of donor 2 CD4+ cells at different time stages showing viability via a Live/Dead stain using Zombie NIR to stain dead/dying cells. Each time point was performed in triplicate cell cultures. The data underlying this figure can be found in the supplemental flow cytometry files uploaded to flowrepository.org.
Figure created with BioRender.com.
(TIFF)

**S3 Fig. Expanded cytokine secretion array in chronically stimulated CD4+ memory cells.**
(A) Immunoblots of conditioned media of 14 cytokines commonly secreted by immune cells. Cells from donor #2 were brought through the exhaustion protocol, and conditioned media was made and collected at day 0, 2, 6, and 12 using methods described for cytokine release assay. Figure created with BioRender.com.
(TIFF)

**S4 Fig. Flow cytometry gating strategy, flow cytometry markers, and co-expression heterogeneity in chronically stimulated CD4+ memory cells.** (A) Example (day 10) for gating strategy used for inhibitory receptor flow cytometry plots in Fig 1. (B) Flow cytometry analysis of LAG-3, CD-137, and CD-160 on day 0, day 6, and day 12 of chronic stimulation time course. (C) tSNE plots showing relative expression across cell population. Different donor was used here, and cells were analyzed on day 11 of exhaustion protocol. The data underlying this figure can be found in the supplemental flow cytometry files uploaded to flowrepository.org.
Figure created with BioRender.com.
(TIFF)

**S5 Fig. Proteomics data quality control and additional plots for CD4+ memory T cell chronic stimulation time course data.** Intensity values of proteins in each sample, before and after loess normalization for (A) first donor and (B) second donor. (C) Venn diagram showing Donor 1 and 2 protein overlap. (D) Venn diagram showing number of proteins identified in

proteomics and phosphoproteomics datasets and their overlap from donor 1. RLE plots for all samples for (E) donor 1 and (F) donor 2. The data underlying this figure can be found in S1 Data (A, E) and S2 Data. Figure created with BioRender.com.
(TIFF)

**S6 Fig. Phosphoproteomics data quality control and additional plots for CD4+ memory T cell chronic stimulation time course data, first donor.** (A) Intensity values of phosphopeptides in each sample, before and after loess normalization. (B) Phosphopeptides mean vs. standard deviation of intensity values. (C) Volcano plots showing differential analysis of day 6 vs. resting, day 8 vs. day 6, day 10 vs. day 8, and day 12 vs. day 10. (D) Global phosphosite expression heatmap with phosphosites clustered based on expression profiles showing pathway enrichment by gene cluster. (E) Number of phosphorylation sites identified sorted by residue. The data underlying this figure can be found in S1 Data. Figure created with BioRender.com.
(TIFF)

**S7 Fig. Expanded flow cytometric analysis of HIV+, HIV-, and combined primary CD4+ T cells showing increased PD-1 expression in CD276+ and FLT-1+ CD4+ T cells.** Mean fluorescence intensity (MFI) of PD-1 in CD4+ T cells from (A) HIV+, (B) HIV-, or (C) both population combined for CD4+ cells that are CD276+ or FLT-1+. The data from this figure can be found in the supplemental flow cytometry files uploaded to flowrepository.org. Figure created with BioRender.com.
(TIFF)

**S8 Fig. Inhibitory receptor and activation marker flow cytometry of p300 inhibited chronically stimulated (day 12) T cells.** Mean fluorescence intensity (MFI) of (A) LAG-3, (B) TIGIT, (C) CD137, (D) CD160, in cells treated with vehicle (DMSO), p300 inhibitor (A-485), Dasatinib, or rested for 2 days. The data underlying this figure can be found in S3 Data. Figure created with BioRender.com.
(TIFF)

**S1 Table. Antibodies used in the PBM microarray.**
(XLSX)

**S2 Table. Antibodies used for FACS.**
(XLSX)

**S1 Data. Omics data summary file.** This file contains the analyzed proteomic, phosphoproteomic, and metabolomic data sets as separate sheets within the excel file. The left columns for each data set are intensity values that have been row Z-scored and have conditional formatting to create a heatmap within excel. Intensity values used for analysis can be found in the rightmost columns, which have been normalized and filtered.
(XLSX)

**S2 Data. Donor 2 proteomics data set.** This file contains the results from the entire proteomic data set for donor 2.
(XLSX)

**S3 Data. Figure numerical values.** This file contains numerical values for any figures shown that cannot be found in the other supplemental materials.
(XLSX)

**S4 Data. Kinase cluster assignments and proteomic data.** Proteomic data and cluster assignments for kinases. Sort by "Cluster" column to see proteins of the same cluster and use the

heatmap (left-side of the sheet) to observe expression patterns.
(XLSX)

**S5 Data. eXpression2Kinases integrative analysis results.** Integrative analysis results from eXpression2Kinases tool. Contains proteins used in the search, as well as all results received and used here. Protein results discussed in this manuscript have been highlighted.
(XLSX)

**S6 Data. Cell surface protein cluster assignments and proteomic data.** Proteomic data and cluster assignments for cell surface proteins. Sort by "Cluster" column to see proteins of the same cluster and use the heatmap (left-side of the sheet) to observe expression patterns.
(XLSX)

**S7 Data. Transcriptional regulators assignments and proteomic data.** Proteomic data and cluster assignments for transcription factors and co-factors. Sort by "Cluster" column to see proteins of the same cluster and use the heatmap (left-side of the sheet) to observe expression patterns.
(XLSX)

**S8 Data. Network analysis file 1.** This file contains the pruned network analysis data displayed in Fig 5B.
(ZIP)

**S9 Data. Network analysis file 2.** This file contains the unpruned network analysis data displayed in Fig 5A.
(ZIP)

**S10 Data. PBM analysis results file.** This file contains the values and results from the PBM analysis shown in Fig 5F.
(TXT)

**S1 File. Gating strategies.**
(ZIP)

## Acknowledgments

All main and supplemental figures were created using BioRender.

## Author Contributions

**Conceptualization:** Matthew L. Lawton, Andrew Emili.

**Data curation:** Matthew L. Lawton.

**Formal analysis:** Matthew L. Lawton, Dante Bolzan, Ahmed Youssef, Dzmitry Padhorny, Dima Kozakov, Stefan Wuchty.

**Funding acquisition:** Andrew Emili.

**Investigation:** Matthew L. Lawton, Yashasvi Tharani.

**Methodology:** Matthew L. Lawton, Melissa M. Inge, Benjamin C. Blum, Weiwei Lin, Jarrod Moore.

**Project administration:** Matthew L. Lawton, Andrew Emili.

**Resources:** Xaralabos Varelas, Gerald V. Denis, Wilson W. Wong, Andrew Emili.

**Supervision:** Andrew Emili.

**Validation:** Matthew L. Lawton, Erika L. Smith-Mahoney, Christina McConney, Jacob Porter, Trevor Siggers, Jennifer Snyder-Cappione.

**Visualization:** Matthew L. Lawton.

**Writing – original draft:** Matthew L. Lawton.

**Writing – review & editing:** Matthew L. Lawton, Andrew Emili.

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
