## [Editor Report · Decision Letter 0]

11 Sep 2024

Dear Dr Emili, 

Thank you for submitting your manuscript entitled "Multiomic Profiling of Chronically Activated CD4+ T Cells Reveals Drivers of T-Cell Exhaustion Phenotype and Metabolic Reprogramming" for consideration as a Research Article by PLOS Biology.

Your manuscript has now been evaluated by the PLOS Biology editorial staff, as well as by the previous academic editor, and I am writing to let you know that we would like to send your submission back to the previous reviewers.

Once your full submission is complete, your paper will undergo a series of checks in preparation for peer review. After your manuscript has passed the checks it will be sent out for review. To provide the metadata for your submission, please Login to Editorial Manager (https://www.editorialmanager.com/pbiology) within two working days, i.e. by Sep 13 2024 11:59PM.

Kind regards,

Melissa

Melissa Vazquez Hernandez, Ph.D.

Associate Editor

PLOS Biology

---

## [Decision Letter · Decision Letter 1]

2 Nov 2024

Dear Dr Emili,

Thank you for your patience while we considered your revised manuscript "Multiomic Profiling of Chronically Activated CD4+ T Cells Reveals Drivers of T-Cell Exhaustion Phenotype and Metabolic Reprogramming" for publication as a Research Article at PLOS Biology. This revised version of your manuscript has been evaluated by the PLOS Biology editors, the Academic Editor and one of the original reviewers.

Based on the reviews, we are likely to accept this manuscript for publication, provided you satisfactorily address the remaining editorial points below. Please also make sure to address the following data and other policy-related requests.

a) We routinely suggest changes to titles to ensure maximum accessibility for a broad, non-specialist readership, and to ensure they reflect the contents of the paper. In this case, we would suggest a minor edit to the title, as follows. Please ensure you change both the manuscript file and the online submission system, as they need to match for final acceptance:

"Multiomic profiling of chronically activated CD4+ T cells identifies drivers of exhaustion and metabolic reprogramming"

b) Please confirm that no financial support was received for this study. 

Please supply the numerical values either in the a supplementary file or as a permanent DOI’d deposition for the following figures:

Figure 2ABCDEFG, 3AB, 4, 5DFE, 6ABC, 7AB, S1B, S3A, S4C, S5ABEF, S6A-E, S7ABC, S8A-D

d) Please cite the location of the data clearly in all relevant main and supplementary Figure legends, e.g. “The data underlying this Figure can be found in S1 Data” or “The data underlying this Figure can be found in https://doi.org/10.5281/zenodo.XXXXX” 

e) Please provide the molecular network files necessary to build Figs 5AB

f) “For figures containing FACS data (Figure S1A, S2, and S4AB), please provide the FCS files and a picture showing the successive plots and gates that were applied to the FCS files to generate the figure. We ask that you please deposit this data in the FlowRepository (https://flowrepository.org/) and provide the accession number/URL of the deposition in the Data Availability Statement in the online submission form.”

g) Please ensure that your Data Statement in the submission system accurately describes where your data can be found and is in final format, as it will be published as written there.

h) Per journal policy, if you have generated any custom code during the course of this investigation, please make it available without restrictions upon publication. Please ensure that the code is sufficiently well documented and reusable, and that your Data Statement in the Editorial Manager submission system accurately describes where your code can be found.

We expect to receive your revised manuscript within two weeks. 

*Published Peer Review History*

*Press*

Sincerely,

Melissa

Melissa Vazquez Hernandez, Ph.D.

Associate Editor

PLOS Biology

REVIEWERS' COMMENTS:

Reviewer #2:

The manuscript entitled "Multiomic Profiling of Chronically Activated CD4+ T Cells Reveals Novel Drivers of T-Cell Exhaustion Phenotype and Metabolic Reprogramming" explores a significant gap in our knowledge of exhaustion. CD4 T cells orchestrate the response of multiple cell types, including dendritic cells, and have a significant role in the overall effectiveness of the immune response. Understanding how CD4 T cell exhaustion influences the immune response against chronic infections and cancer is of vital importance for increasing our ability to intervene and improve patient outcomes. The authors have provided a lot of additional work to validate their original findings and have significantly improved the quality and relevance of the work presented. The authors have adequately addressed both reviewer 1 and reviewer 2's comments or have provided appropriate explanations of why the experiments was not feasible/should be part of a follow up paper. I only have one minor comment remaining and that is to added error bars to figure 1D.

---

## [Editor Report · Decision Letter 2]

15 Nov 2024

Dear Dr Emili,

Thank you for the submission of your revised Research Article "Multiomic profiling of chronically activated CD4+ T cells identifies drivers of exhaustion and metabolic reprogramming" for publication in PLOS Biology. On behalf of my colleagues and the Academic Editor, Avinash Bhandoola, I am pleased to say that we can in principle accept your manuscript for publication, provided you address any remaining formatting and reporting issues. These will be detailed in an email you should receive within 2-3 business days from our colleagues in the journal operations team; no action is required from you until then. Please note that we will not be able to formally accept your manuscript and schedule it for publication until you have completed any requested changes.

PRESS

Sincerely, 

Melissa

Melissa Vazquez Hernandez, Ph.D., Ph.D.

Associate Editor

PLOS Biology
